# Bladder cancer therapy using a conformationally fluid tumoricidal peptide complex

Antonín Brisuda[1,7], James C. S. Ho [2,6,7], Pancham S. Kandiyal[3], Justin T-Y. Ng[4], Ines Ambite [2], Daniel S. C. Butler [2], Jaromir Háček[5], Murphy Lam Yim Wan[2], Thi Hien Tran[2], Aftab Nadeem[2], Tuan Hiep Tran[2], Anna Hastings [3], Petter Storm[2], Daniel L. Fortunati[3], Parisa Esmaeili[2], Hana Novotna[1], Jakub Horňák[1], Y. G. Mu [4], K. H. Mok [3], Marek Babjuk[1,8] & Catharina Svanborg [2,8✉]

Partially unfolded alpha-lactalbumin forms the oleic acid complex HAMLET, with potent tumoricidal activity. Here we define a peptide-based molecular approach for targeting and killing tumor cells, and evidence of its clinical potential (ClinicalTrials.gov NCT03560479). A 39-residue alpha-helical peptide from alpha-lactalbumin is shown to gain lethality for tumor cells by forming oleic acid complexes (alpha1-oleate). Nuclear magnetic resonance measurements and computational simulations reveal a lipid core surrounded by conformationally fluid, alpha-helical peptide motifs. In a single center, placebo controlled, double blinded Phase I/II interventional clinical trial of non-muscle invasive bladder cancer, all primary end points of safety and efficacy of alpha1-oleate treatment are reached, as evaluated in an interim analysis. Intra-vesical instillations of alpha1-oleate triggers massive shedding of tumor cells and the tumor size is reduced but no drug-related side effects are detected (primary endpoints). Shed cells contain alpha1-oleate, treated tumors show evidence of apoptosis and the expression of cancer-related genes is inhibited (secondary endpoints). The results are especially encouraging for bladder cancer, where therapeutic failures and high recurrence rates create a great, unmet medical need.

[1] Department of Urology, Motol University Hospital, 2nd Faculty of Medicine, Charles University Praha, Prague, Czech Republic. [2] Division of Microbiology, Immunology and Glycobiology, Department of Laboratory Medicine, Faculty of Medicine, Lund University, Lund, Sweden. [3] Trinity Biomedical Sciences Institute (TBSI), School of Biochemistry & Immunology, Trinity College Dublin, The University of Dublin, Dublin, Ireland. [4] School of Biological Sciences, Nanyang Technological University, Singapore, Singapore. [5] Department of Pathology and Molecular Medicine, Motol University Hospital, 2nd Faculty of Medicine, Charles University Praha, Prague, Czech Republic. [6] Present address: Centre for Biomimetic Sensor Science, Nanyang Technological University, Singapore, Singapore. [7] These authors contributed equally: Antonín Brisuda, James C.S. Ho. [8] These authors jointly supervised this work: Marek Babjuk, Catharina Svanborg. ✉email: catharina.svanborg@med.lu.se

Targeted cancer therapies have made significant advances but the lack of tumor specificity remains a significant concern[1,2]. Few current therapies kill cancer cells without harming healthy tissues, and severe side effects have become accepted as a necessary price to pay for survival or cure. The notion of successfully combining efficacy with increased tumor selectivity is justly regarded with skepticism. Yet, a serendipitous discovery has provided insights into mechanisms of tumor-specific cell death, induced by unfolded polypeptide chains, which acquire tumoricidal activity by forming fatty acid complexes[3,4]. Extensive observations in tumor models and clinical studies have further defined the protein–lipid complexes as an interesting class of molecules with significant therapeutic potential[5].

The findings are challenging, as protein unfolding and loss of structural definition is associated with a gain of toxicity, due to the formation of amorphous aggregates and amyloid fibrils[6,7]. Native protein structure is often regarded as a prerequisite for biological function, by epitope-specific interactions and molecular fitness for a finite number of cellular targets. Yet, a lack of structural definition may, in some cases, result in a gain of function, in part by uncovering different conformations and exposing peptide motifs that are unavailable in the native state[8,9]. Such effects have been predicted for membrane perturbing α-helices in antimicrobial peptides, where the ability to destabilize lipid bilayers has been proposed to reside in the three-dimensional conformation rather than the amino acid sequence[10].

Alpha-lactalbumin is crucial for the survival of lactating mammals. In its native state, the protein serves as a substrate specifier in the lactose-synthase complex[11], defining the nutritional content and fluidity of milk. Partially unfolded alpha-lactalbumin, in contrast, forms an oleic acid complex, named HAMLET, with potent tumoricidal activity[3,4,8,9]. The HAMLET complex kills a range of tumor cells with rapid kinetics and shows therapeutic efficacy in animal models of colon cancer, glioblastoma, and bladder cancer[12–15]. Early, investigator-driven clinical studies demonstrated that HAMLET is active topically, against skin papilloma and induces tumor cell shedding into the urine in patients with bladder cancer[5,16].

This study presents a synthetic, peptide-based drug candidate derived from alpha-lactalbumin, which reproduces the tumoricidal properties of HAMLET and allows for a full translation of these findings into the clinic. Through complementary nuclear magnetic resonance (NMR) analysis and computational modeling, the molecular basis for this "gain-of-function" is defined, including the three-dimensional structural motifs that determine fatty acid-binding efficiency and tumoricidal activity. The therapeutic efficacy of the complex is demonstrated in patients with non-muscle invasive bladder cancer (NMIBC), in a fully controlled clinical trial.

## Results

### Peptide-specific tumoricidal activity

To understand the involvement of specific peptide motifs in tumor cell death, we synthesized the N-terminal alpha-helical domain (residues 1–39, alpha1) or the beta-sheet (40–80, beta) domains of human alpha-lactalbumin (Fig. 1a). The alpha1 peptide formed complexes with oleate (alpha1–oleate, 1:5) and circular dichroism (CD) spectra detected an increase in alpha-helical structure content in these complexes (Fig. 1b). The beta–oleate complex remained structurally unchanged (Fig. 1b). Alpha1–oleate triggered a rapid, dose-dependent death response in human lung- and kidney carcinoma cells and in murine bladder cancer cells (Fig. 1c, d). The beta–oleate complex lacked tumoricidal activity and tumor cells subjected to the naked alpha-helical peptides (35 μM) or oleate (175 μM) controls were not tumoricidal (Fig. 1c, d and Supplementary Fig. 1). The loss of cell viability was irreversible, as shown after 10 days, by using colony assays (Fig. 1e and Supplementary Fig. 1). Membrane blebbing occurred in tumor cells within minutes of exposure to alpha1–oleate but the naked peptide- and oleate controls were not active (Fig. 1f and Supplementary Fig. 1). Rapid $K^+$ fluxes were recorded, further defining the membrane response (Fig. 1g). Pretreatment of the cells with $Na^+$ and $K^+$ flux inhibitors reduced cell death by 40–50%, linking the membrane response to tumor cell death (Fig. 1h). The alpha1–oleate complex was rapidly internalized by tumor cells and by TUNEL staining, alpha1–oleate was shown to induce double-strand DNA breaks in the tumor cells, indicative of apoptosis (Fig. 1i, j).

In a screen of proteins with membrane-integrating properties, SAR1 was found to form tumoricidal complexes with oleic acid, reproducing effects of alpha1–oleate (Supplementary Figs. 1 and 2). SAR1 is a membrane-integrating protein of the COPII complex that induces membrane tubulation by insertion of its N-terminal amphipathic α-helix[17–19]. The N-terminal alpha-helical peptide (residues 1–23, sar1alpha) formed an oleate complex, which efficiently killed tumor cells (Supplementary Figs. 1 and 2). The naked peptide- and oleate controls were not active (Supplementary Fig. 1). Sar1alpha–oleate triggered membrane blebbing in tumor cells, rapid $K^+$ fluxes were recorded, and tumor cell death was partially inhibited by $Na^+$ and $K^+$ flux inhibitors, suggesting a similar mode of action of the two complexes, despite low sequence homology (Supplementary Fig. 2). The sar1beta–oleate complexes and naked peptide controls did not trigger tumor cell death, however (Supplementary Figs. 1 and 2).

In preparation for the clinical trial, the safety of alpha1–oleate was investigated in C57BL/6J mice carrying MB49-induced bladder tumors[13]. Therapeutic efficacy in 100% of treated mice compared to the sham group and a lack of toxicity was demonstrated, providing the necessary background to plan the clinical trials[13].

### Biomolecular NMR analysis of the peptide–oleate complexes

$^1H$ NMR spectra of the alpha1–oleate and sar1alpha–oleate complexes detected a shift from sharp signals for the naked peptides to broad signals and poor chemical shift dispersion for the oleate complexes (Fig. 2a, b), suggesting a conformational change from a random-coil fast-exchange time regime to an intermediate millisecond timescale. Broadening in the amide, side-chain methyl and aromatic regions suggests that interactions between fatty acids and peptides occur throughout the molecules. Two-dimensional nuclear Overhauser effect spectroscopy (2D NOESY) spectra identified non-covalent, relatively short through-space interactions between the respective peptides and fatty acids. Important nuclear Overhauser effects (NOEs) were detected between the olefinic protons (5.23 ppm) of oleic acid and the Hα and aromatic protons of alpha1 and between the sar1alpha aromatic region and the oleic acid olefinic protons (Fig. 2c, d). The downfield chemical shift of amide protons observed between 7.6 and 8.8 ppm suggests the presence of secondary structure in alpha1 and alpha1–oleate. Well-resolved signals obtained from the one-dimensional $^1H$ NMR spectra provided a stoichiometry of 3.7 oleate molecules per alpha1 peptide. Chemical shift mapping revealed a cluster of residues with aliphatic side chains that change upon the binding of oleate, providing further evidence of interactions between peptides and fatty acids (Fig. 2e, f).

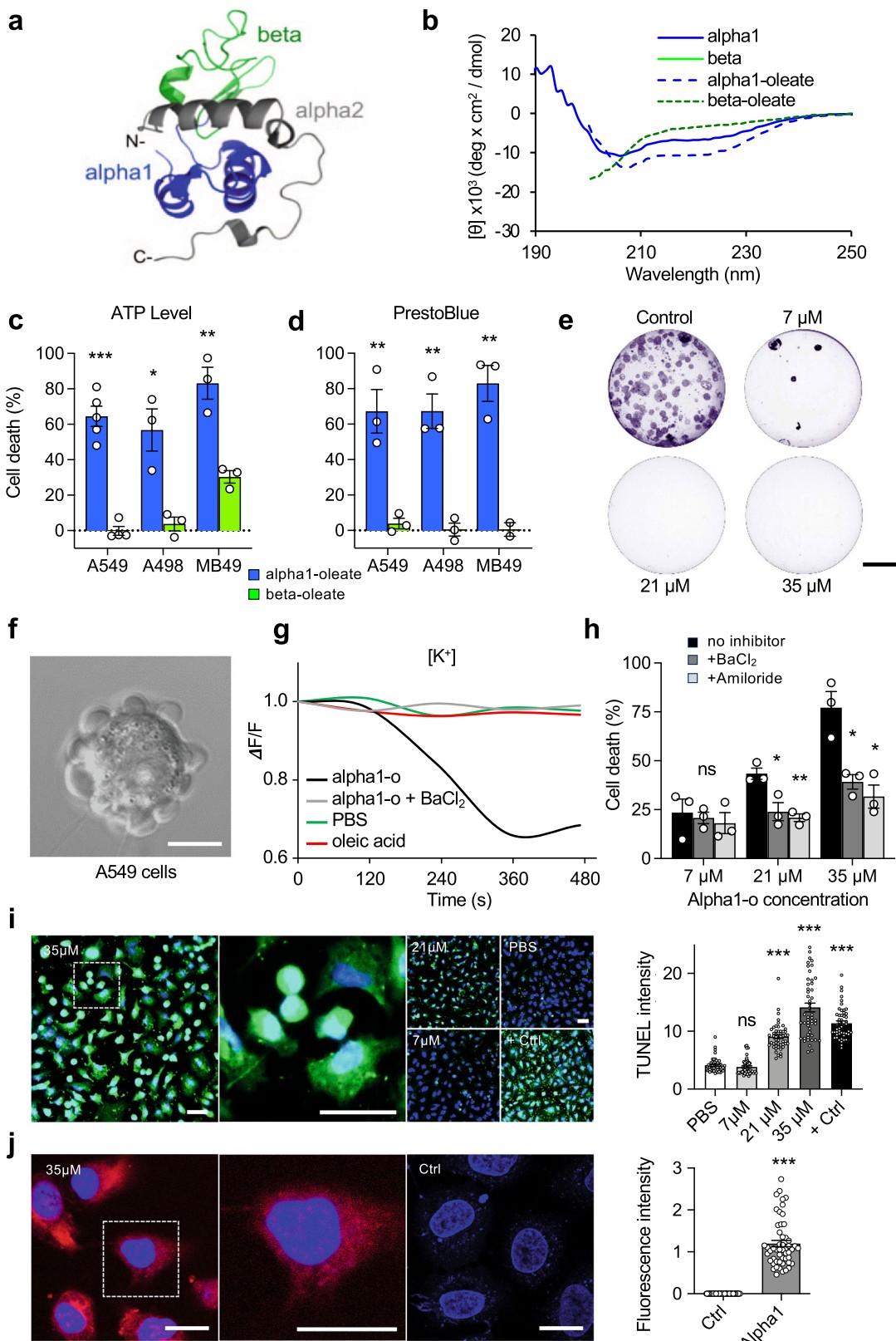

Hydrodynamic volume measurements carried out with size-exclusion high-performance liquid chromatography (SE-HPLC) and diffusion-ordered NMR spectroscopy (DOSY) showed that the alpha1–oleate complex $R_H$ was considerably larger (27.4 and 29.3 Å, respectively) than the naked peptides (16.1 Å from SE-HPLC and 14.4 Å from DOSY) (Fig. 2g, h and Supplementary

Figs. 3 and 4). The distinctly larger $R_2$ values (transverse relaxation rate) for the complex than those of alpha1 peptide and human serum albumin (HSA) suggested slower millisecond to microsecond exchange processes (Supplementary Table 1 and Supplementary Figs. 5 and 6). Importantly, the $R_2$ values for the complex were also different from oleate in an aqueous solution,

**Fig. 1 Tumoricidal activity of two non-homologous alpha-helical peptide–oleate complexes. a** Ribbon representation of the crystallographically determined three-dimensional structure of human α-lactalbumin (PDB ID: 1B9O), indicating the alpha1 (blue), beta (green), and alpha2 (gray) domains. The calcium ion is not shown. **b** Far-UV circular dichroism spectra of synthetic alpha1 peptide, beta peptide, and their respective peptide–oleate complexes. **c, d** Death response in human lung (A549), kidney (A498), and murine bladder (MB49) carcinoma cells, quantified as a reduction in ATP levels (**c**, $P =$ 3.26E−5 for A549, 0.013 for A498 and 0.005 for MB49, alpha1–oleate compared to beta–oleate) or PrestoBlue fluorescence (**d**, $P = 0.007$ for A549, 0.003 for A498 and 0.002 for MB49, alpha1–oleate compared to beta–oleate). Cells were treated with the alpha1–oleate complex (blue) or the beta–oleate complex (green), (3 h, 35 μM, cell death compared to PBS controls). For controls exposed to the naked peptides or oleate alone, see Supplementary Fig. 1d. **e** Colony assay showing dose-dependent long-term effects of alpha1–oleate. A representative image is shown from two independent experiments. Scale bar = 5 mm. **f** Alpha1–oleate triggers rapid membrane blebbing in A549 lung carcinoma cells (35 μM, 10 min). Scale bar = 10 μm. A representative image is shown from three independent experiments. **g** $K^+$ efflux in A549 lung carcinoma cells exposed to alpha1–oleate and inhibition with $BaCl_2$. **h** Inhibition of cell death by the ion flux inhibitors Amiloride and $BaCl_2$ (100 μM), measured by PrestoBlue fluorescence ($P = 0.031$ for 21 μM + $BaCl_2$, 0.005 for 21 μM + Amiloride, 0.028 for 35 μM + $BaCl_2$, and 0.014 for 35 μM + Amiloride, compared to no inhibitor). **i** DNA strand breaks detected by TUNEL staining in alpha1–oleate-treated A549 lung carcinoma cells ($n = 50$ cells per group). Scale bar = 20 μm. **j** AlexaFluor568-labeled alpha1–oleate (red) is internalized by A549 lung carcinoma cells. Nuclei are counterstained with DAPI (blue) ($n = 52$ cells per group). Scale bar = 10 μm. Data are presented as mean ± SEM from three independent experiments, *$P < 0.05$, **$P < 0.01$, ***$P < 0.001$, analyzed by two-tailed unpaired $t$-test (**c, d, h, j**) and 2-way ANOVA using Dunnett's correction (**i**).

suggesting that the dynamics of the complex were clearly different from micelle/vesicle-like particles formed from oleic acid/oleate with no peptide component.

**Computational analyses of the peptide- and peptide–oleate system**. Computational simulations also pointed toward structural heterogeneity, showing that the naked peptides and peptide–oleate complexes belonged to wide conformational spaces with relatively deep basins (Fig. 3a, b). Representative structures mapped to different free energy surface minima, revealed prominent alpha-helical secondary structural elements and a hydrophobic oleate core for the peptide–oleate complexes (Fig. 3c, e). The peptide–oleates fold upon this core differently from the naked peptides, which exhibit multiple local minima (Fig. 3d, f, and Supplementary Tables 2 and 3). Naked alpha1 ensembles were characterized by various partially folded helix-turn conformations, whereas naked sar1alpha ensembles exhibited a mixture of the random coil, alpha-helical, and beta-sheet structures.

A contact probability analysis revealed that the interactions between alpha1 or sar1alpha and oleate were mainly hydrophobic, with a >0.9 contact probability with olefinic protons (Supplementary Tables 4 and 5). The peptide–oleate complexes displayed relatively wide and deep free energy minima basins, suggesting that a multitude of confirmations would be equally possible to visit (Fig. 3c, e). When combined with the $R_2$ relaxation rates, the possibility of multiple sampling of various conformations within a short period of time provides an argument that rather than targeting specific partners, these alpha-helical complexes may potentially be interacting with multiple putative binding partners available on cancer cell surfaces[20].

Based on these extensive investigations and the strong agreement of the experimental aspects with the simulated predicted ensembles, it was clear that the apparently unrelated peptides alpha1 and sar1alpha can form complexes with shared structural characteristics, involving a flexible peptide moiety and a fatty acid cluster. This notion resonates with the sequence-function inconsistency among certain antimicrobial amphipathic alpha helices, where peptides with similar overall properties, such as hydrophobicity or charge, can have dramatically different levels of activity[10].

**A placebo-controlled, randomized clinical trial of alpha1–oleate in patients with NMIBC**. NMIBC is common and despite current treatment protocols, recurrence rates are high[21,22]. To address if the therapeutic effects observed in the murine MB49 bladder

cancer model can be translated into the clinic, the investigational product alpha1–oleate was produced under GMP conditions. The alpha1–oleate complex was further subjected to formal toxicity testing and the results have been published[13]. Toxicity for bladder tissue was not detected at concentrations ranging from 1.7 to 17 mM[13].

The clinical safety and therapeutic potential of alpha1–oleate were tested in a single-center, placebo-controlled, double-blind Phase I/II trial (EudraCT No: 2016-004269-14, ClinicalTrials.gov NCT03560479, Supplementary Note 1, Supplementary Table 6). Patients with suspected NMIBC were randomized 1/1 to receive alpha1–oleate or placebo during a period of 22 days, prior to endoscopic removal of the tumor by transurethral resection (TURB), (Fig. 4a, b). Alpha1–oleate (1.7 mM) or placebo (PBS) was administered intravesical on six occasions (30 mL, days 1, 3, 5, 8, 15, and 22). The placebo solution was identical in appearance to the active treatment. Demographic data, medical history, and vital signs did not differ between the treatment and placebo groups (for details see Supplementary Table 7).

**Primary study endpoints**. Adverse events (AEs) were recorded and coded according to MedDRA (version 21.1) with a safety follow-up after 52 days (Supplementary Table 8). Procedure-related AEs, such as dysuria and bacteriuria, occurred at a similar rate in the treatment and placebo groups. AEs specific for the treatment group were not detected, suggesting low toxicity of the study medication (Fig. 4c). Furthermore, there was no evidence of a toxic response in healthy tissue samples from patients treated with alpha1–oleate, defined by histopathology or TUNEL staining.

Tumor cell shedding and release of tumor cell clusters were recorded. Cells with uroepithelial morphology were quantified in urine at each visit, before instillation and about 2 h after the instillation of alpha1–oleate or placebo. Alpha1–oleate triggered a rapid increase in cell shedding compared to the pre-instillation sample in all treated patients, at all visits (Fig. 5a–c and Supplementary Fig. 7). In addition, tumor cell clusters were released into the urine in the treatment group. The clusters were relatively large and the presence of tumor stroma in some samples supported their tumor origin (Fig. 5d–f). The cells shed in urine were assigned a pathology score as per the Paris classification (classes 1–6, urine cytology was a secondary endpoint). In the treatment group, an increase in the Paris score was detected in post-inoculation samples compared to pre-inoculation samples (Fig. 5g). A low level of cell shedding in the placebo group was attributed to the instillation procedure and changes in pathology

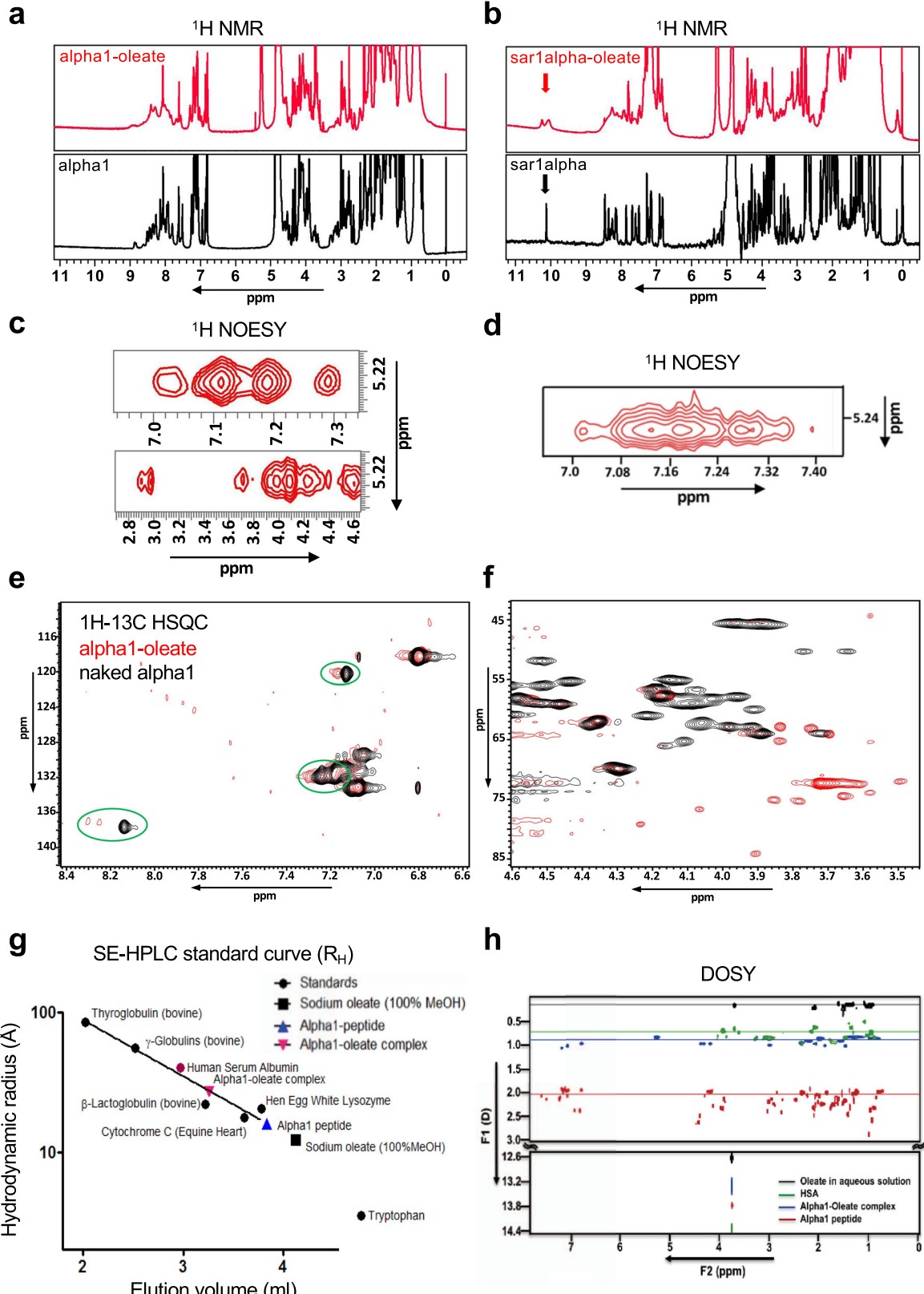

score or shedding of cell clusters were not observed (Fig. 5a–g and Supplementary Fig. 7).

The endoscopic appearance of the tumors was recorded at the time of diagnosis and prior to surgery using a flexible cystoscope with white light-band and narrow-band imaging. Sizes were assessed by an experienced endourologist using fully opened

clamps of the flexible forceps and a measuring device close to the tumor. Paired images from 39 patients were evaluated in a blinded manner, by an independent NMIBC expert using a simplified Delphi method[23] addressing changes in lesion size, superficial necrosis, and tissue vascularization. A reduction in lesion size was detected in the treatment group ($n = 19$, Fig. 5h, i).

**Fig. 2 Biomolecular NMR analysis of naked alpha1- and sar1alpha peptides and their oleate complexes. a, b** One-dimensional $^1$H NMR spectra. The naked alpha1- (**a**, black) and sar1alpha- (**b**, black) peptides assume an ensemble of structures that interconvert rapidly and are therefore seen as sharp peaks. The alpha1–oleate (**a**, red) and sar1alpha–oleate complexes (**b**, red) show broader peaks. Arrows indicate the indole $^1$H signals arising from the three Trp side chains present in the sar1alpha peptide. **c, d** Two-dimensional $^1$H NOESY spectra of alpha1–oleate and sar1alpha–oleate complexes, showing atomic-level proximities of the fatty acid to the respective peptide. The spectra highlight NOEs between the 9,10 olefinic protons (5.23 ppm) of oleic acid with the Hα protons and aromatic protons of the alpha1–oleate complex (**c**) and the sar1alpha–oleate complex (**d**). **e, f** Two-dimensional $^1$H–$^{13}$C Heteronuclear Single Quantum Coherence (HSQC) spectra overlays of the alpha1 peptide (red) and alpha1–oleate complex (black). Chemical shift perturbation is detected in the aromatic side-chain region and the imidazole ring protons (**e**, green circled regions), and in the aliphatic side chain regions (**f**). **g** Size-exclusion HPLC (SE-HPLC) of the alpha1 peptide and the alpha1–oleate complex, mapped onto a standard calibration curve. **h** Diffusion-ordered NMR spectroscopy (DOSY) of the alpha1 peptide, alpha1–oleate complex, human serum albumin (HSA), and oleate suspension.

In 1 patient, paired pre-treatment and post-treatment images could not be collected for technical reasons. No difference in superficial necrosis or vascularization was observed and there was no change in lesion size in the placebo group ($n = 20$).

**Secondary end points**. Evidence of tumor cell apoptosis was obtained by staining of tumor biopsies obtained at the time of surgery. Biopsies were examined for evidence of treatment-induced apoptosis, quantified by the TUNEL assay (Fig. 6a–c and Supplementary Fig. 8). A significant increase in net mean fluorescence was detected in the treatment group, compared to placebo (see also Fig. 1). Staining was most intense adjacent to the lumen suggesting that a gradient might be formed, from the lumen towards the center of the tumor. TUNEL staining intensity was significantly correlated to cell shedding and alpha1–oleate uptake in individual patients (Fig. 6d) but not to the tumor grade. In healthy tissue biopsies from the treated patients, TUNEL staining was low. Tumors from the placebo group did not show increased TUNEL staining, suggesting that tumor cell apoptosis may be induced by the alpha1–oleate treatment (Fig. 6c).

The alpha1–oleate content of shed cells in urine was quantified by immunohistochemistry, using alpha1-specific antibodies. Alpha1-staining was detected in 70% of post-inoculation samples in the treatment group (Fig. 6e, f and Supplementary Fig. 9). Uptake correlated with cell shedding and cluster grade but not with the tumor grade or stage (Supplementary Fig. 9).

The response to alpha1–oleate was further evaluated by RNA-seq, using RNA from tumor biopsies and comparing the treatment to the placebo group. A strong treatment effect was detected (Fig. 7a–c and Supplementary Fig. 10). Cancer-related genes accounted for about 80% of the significantly regulated genes in the treated patients (cut off fold change > 2.0, $P < 0.05$), confirming the effect of alpha1–oleate on the tumor environment. Genes regulating tumor growth and invasion were inhibited and Ras signaling was suppressed, consistent with known effects of the complex on Ras family members[24] (Fig. 7d and Supplementary Fig. 10). Bladder cancer genes were specifically regulated, including metalloproteinases, solute carriers, WNT complex constituents, and thrombospondin, which affects angiogenesis[25] (Fig. 7e). Furthermore, Fatty Acid Desaturase 6 (*FADS6*) and transcriptional activator *CREB3L4* were affected, suggesting that the tumors respond to the constituents of the alpha1–oleate complex. *FADS6* regulates oleate biosynthesis and *CREB3L4* the unfolded protein response to conformationally fluid proteins, such as the alpha1 peptide. Interesting targets also included the gap junction alpha1 protein, which was inhibited, potentially promoting cell detachment (*GJA1/CXA1* encoding Connexin 43, Supplementary Fig. 10). No difference in tumor grade was observed between the treatment or placebo groups, defined by WHO 1973 and 2004/2016 criteria (Supplementary Table 9). Data regarding two secondary end-points are not reported. As this is an interim analysis, the long-term treatment effects will be evaluated when the entire study has been completed. The urine proteomics data set has not been fully analyzed.

## Discussion

Bladder cancer is the fourth most common malignancy in the United States and the fifth in Europe, with a prevalence of about 1/4000[26]. Due to high recurrence rates and a lack of curative therapies, "bladder cancer has the highest lifetime treatment costs per patient of all cancers, followed by colorectal-, breast-, prostate-, and lung cancer"[27]. More than 80% recur after complete surgical removal of the first tumor and 15% progress to muscle-invasive disease[28]. Intravesical chemotherapy and Bacillus Calmette–Guérin (BCG) immunotherapy have limited efficacy and significant side effects[29,30]. Systemic administration of PD-1 and PD-L1 inhibitors is considered only in BCG unresponsive patients where the experience is limited. Therapeutic options are also limited by the inadequate supply of immunotherapy and chemotherapy drugs worldwide[31]. In this study, we identify conformationally fluid peptide–fatty acid complexes as additional tools in cancer therapy and show that intra-vesical inoculation of alpha1–oleate is safe and effective in patients with bladder cancer.

The tumor response to alpha1–oleate was analyzed in-depth, using cellular and molecular tools to detect changes induced by the complex. Treatment triggered the shedding of cells and tissue fragments into the urine and alpha1–oleate internalization by tumor cells confirmed the affinity of the complex for the tumor. Further analysis of tissue biopsies suggested a lasting effect of the alpha1–oleate instillations, as several tumor samples showed a gradient-like pattern of apoptosis, starting from the bladder lumen. Dysfunctional apoptosis has been identified as a key to tumor development, especially in environments where oncogenes such as MYC drive tumor cell proliferation[32]. Numerous attempts have been made to develop apoptosis-inducing therapeutics with tumor selectivity, but this has proven challenging, probably due to the heterogeneity of individual tumors as well as their intrinsic resistance to activating cell death pathways. The ability of alpha1–oleate to stimulate apoptosis in the majority of bladder tumors is, therefore, encouraging and consistent with the apparent lack of toxicity for bladder tissue.

RNA sequencing revealed profound molecular changes in treated tissues, attributable to alpha1–oleate. Classical cancer gene networks were strongly inhibited in the treated patients, compared to the placebo group, including Ras, previously identified as a target for HAMLET; the oleate complex formed by the alpha-lactalbumin holoprotein[24]. HAMLET binds activated Ras at the plasma membrane of tumor cells and inhibits the Ras signaling pathway, in part through effects on b-Raf phosphorylation. Significant effects on adaptive immunity were not detected and innate immunity was largely inhibited, including granulocyte activation pathways. Notably, genes involved in oleate metabolism and the unfolded protein response were affected, possibly reflecting a direct response to the constituents of the alpha1–oleate complex. In addition, treatment inhibited *GJA1*, a

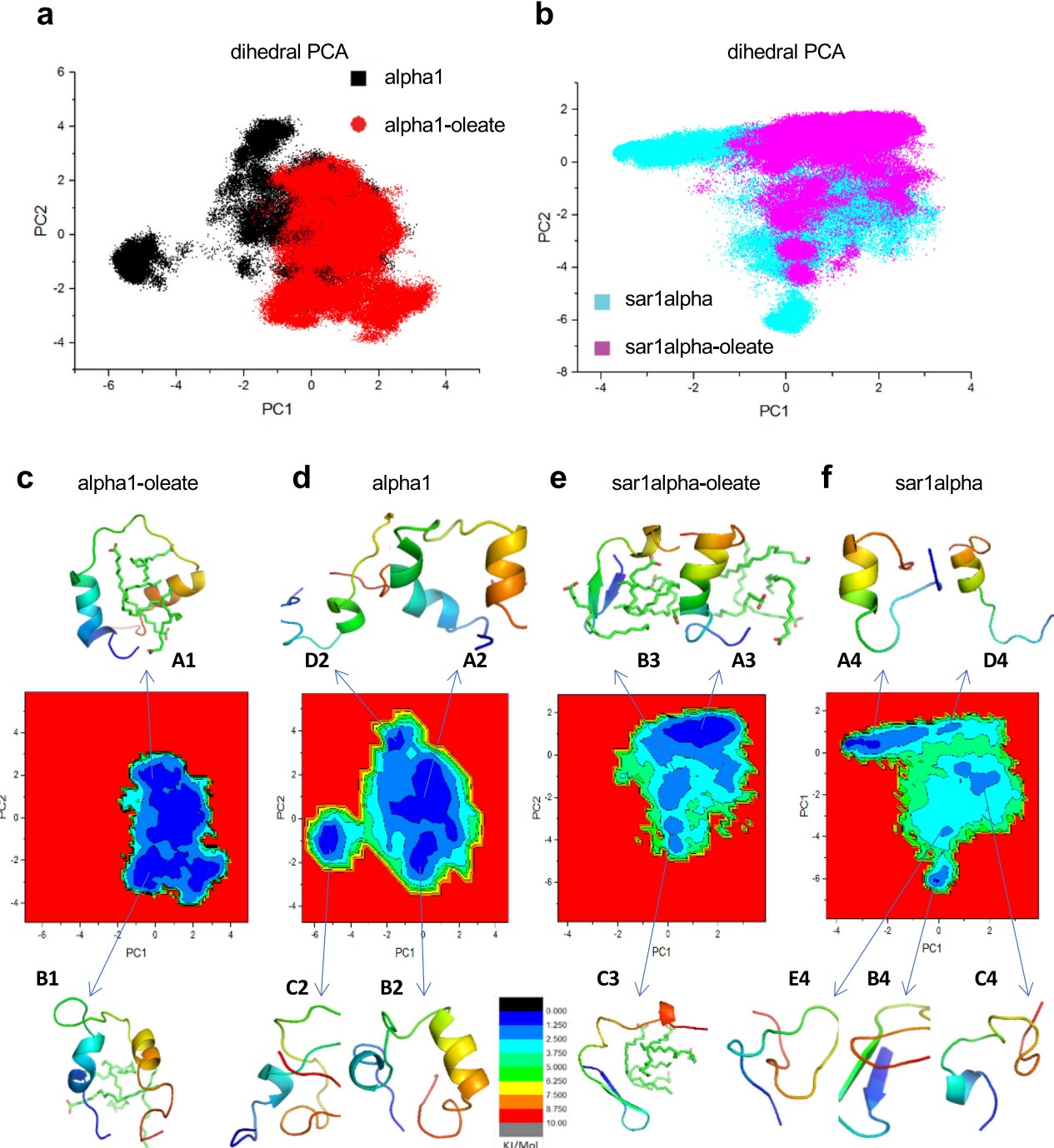

**Fig. 3 Free energy surface analyses of the peptide- and peptide–oleate system. a, b** Superimposition of dihedral principal component analysis (PCA) plots of alpha1 (black points) and alpha1–oleate (red points) systems (**a**), and sar1alpha (cyan points) and sar1alpha–oleate (magenta points) systems (**b**). Principal component (PC)1 and PC2 represent the axes of the two greatest variances after mathematical transformation for dimension reduction. **c–f** Free-energy surfaces as a function of the first two principal components for alpha1–oleate (**c**), naked alpha1 (**d**), sar1alpha–oleate (**e**), and naked sar1alpha (**f**). The representative structures of peptides or peptide–oleate complexes, along with their respective local minima annotations, are colored from the N termini (blue) to the C termini (orange/brown). The free-energy surface of the alpha1–oleate complex contains 2 minima basins, A1 and B1, with A1 representing the major conformational ensemble. The free-energy surface of the sar1alpha–oleate complex contains 3 minima basins, A3, B3, and C3 (with the A3 basin harboring the major structural ensemble), which are characterized by a prominent alpha-helical secondary structural element, as shown from simulation calculated alpha-helical propensities. By contrast, the free-energy surface of the naked sar1alpha shows large structural heterogeneity. Here, minima basins A4 and D4 are represented by helical structures, B4 by beta structure, and C4 and E4 by random coil structures.

gap junction protein that has been proposed to promote cancer development and metastasis[33,34]. The effect occurred specifically in tumor tissue, potentially providing the alpha1–oleate complex with a mechanism to trigger cell shedding, as observed here. It is interesting to speculate that cell shedding may serve as a "tip-of the iceberg" marker of the profound changes in tumor biology that include activation of programmed cell death, transcriptional reprogramming, and inhibition of tumor progression.

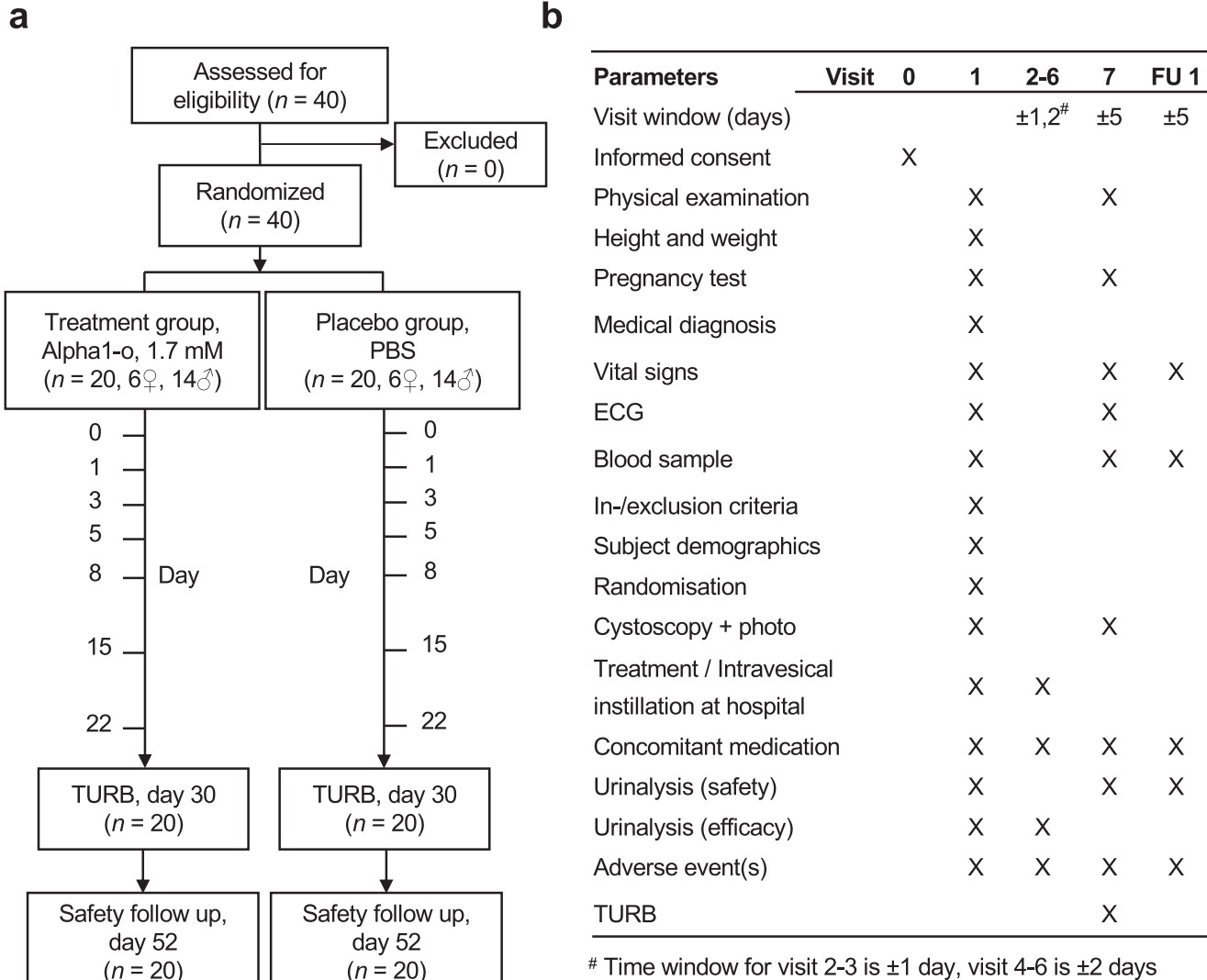

**Fig. 4 Clinical study protocol, demographic data, and adverse events. a** Study CONSORT diagram. **b** Study protocol. After diagnosis and informed consent, the subjects received intravesical instillations of either alpha1–oleate or placebo on six occasions during one month preceding a scheduled transurethral resection (TURB). A safety follow-up was performed 52 days after the first instillation. **c** Number of adverse events (AEs) in the active and placebo groups. No drug-related adverse events were recorded. There were totally 29 AEs reported by 12 subjects in the active group and by 11 subjects in the placebo group. None of the AEs were related to the investigational product. One AE was severe and two were moderate in the placebo group. The active group had one moderate AE. Two subjects in the placebo group reported severe AE (SAEs). The AEs were evaluated descriptively, and the AE profiles were similar between the placebo and the active groups.

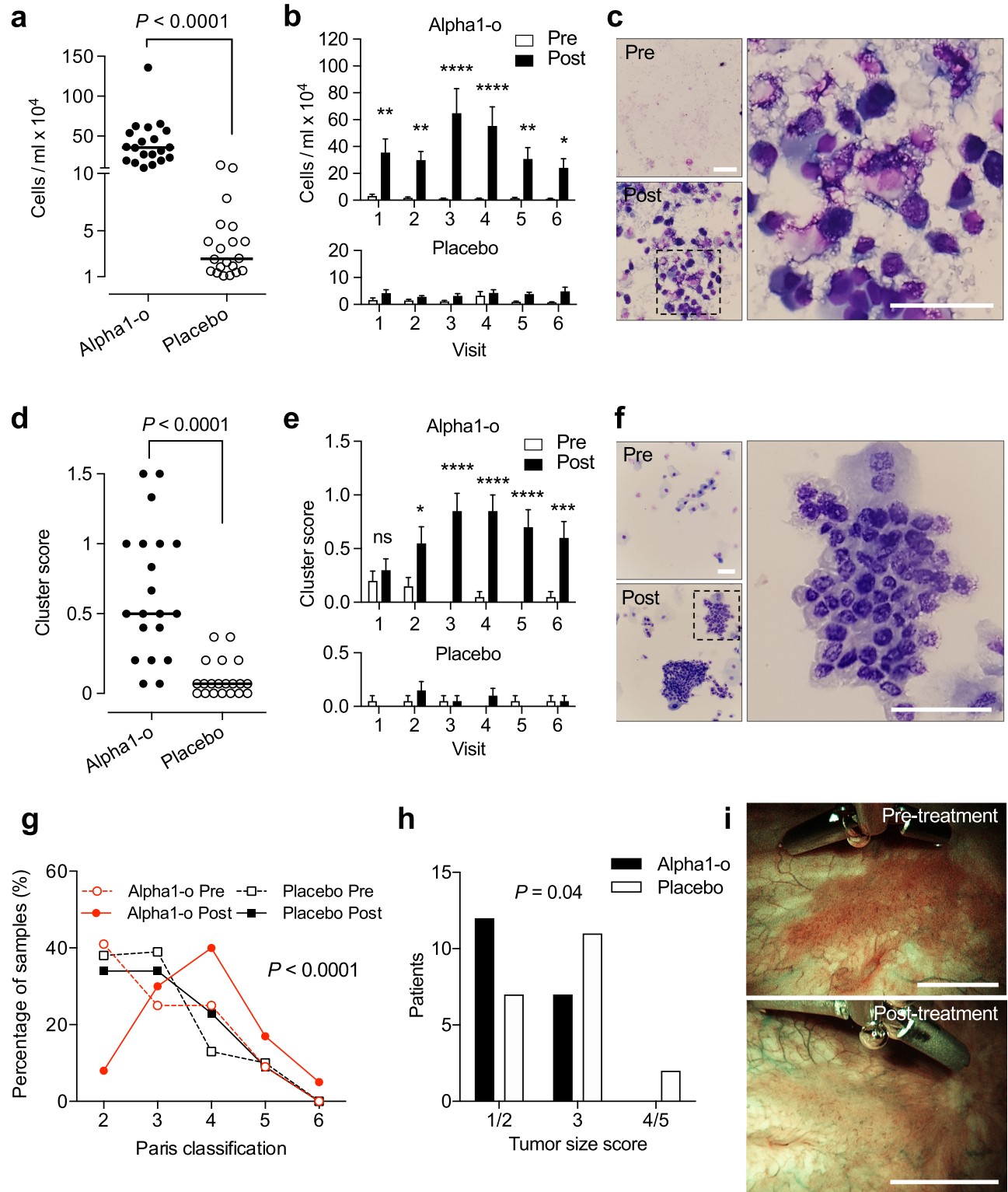

Alternative therapeutic tools are actively being developed and tested in patients with NMIBC, particularly in patients with disease recurrence after BCG treatment[35–37]. Device-assisted hyperthermia was shown to increase the efficacy of intra-vesical chemotherapy but treatment was accompanied by side effects, reducing compliance[38–40]. An oncolytic-virus-based intra-vesical therapy was recently reported to achieve a complete response in 53.4% of patients with BCG-unresponsive carcinoma in situ, in a phase III trial[41]. The authors discuss the assessment of side effects and the development of biomarkers to help select patients suitable for this therapy. In patients with BCG unresponsive disease, treated with systemic Pembrolizumab, a 41% response rate was reported but side effects were prevalent, limiting compliance (Keynote-676 trial[35]). The present study identifies alpha1–oleate as an active drug candidate with low toxicity. Further dose-finding clinical studies and adjuvant therapy protocols will be essential to define the therapeutic window of this complex.

**Fig. 5 Primary endpoints: shedding of tumor cells and reduction in tumor size following intra-vesical instillation of alpha1–oleate. a–c** Cell shedding increased significantly after alpha1–oleate instillation. **a** Scatterplot showing individual means of six visits per patient in the treatment group ($n = 20$) compared to patients receiving placebo ($n = 20$). Line represents the median. **b** Comparison of cell numbers in urine before (pre = white) and after (post = black) alpha1–oleate inoculation on visits 1–6 showing increased cell numbers post-inoculation in the treatment group ($n = 20$ patients per group, $P = 0.0030$ for visit 1, 0.0098 for visit 2, <0.0001 for visits 3 and 4, 0.0073 for visit 5 and 0.0336 for visit 6) but not in the placebo group. Data are presented as mean ± SEM. **c** Representative images, illustrating the increase in cell shedding after alpha1–oleate instillation. Magnification = ×400. Scale bar = 50 μm. **d–f** Difference in the shedding of tumor cell clusters between the treatment and placebo group. **d** Scatterplot showing individual means of six visits per patient in the treatment group compared to patients receiving placebo. Line represents the median. **e** Increased numbers of cell clusters in post-inoculation samples of patients receiving alpha1–oleate ($n = 20$ patients per group, $P = 0.9743$ for visit 1, 0.0212 for visit 2, <0.0001 for visits 3, 4, and 5, and 0.0005 for visit 6). Data are presented as mean ± SEM. **f** Representative images of cancer cell clusters after alpha1–oleate instillation. Magnification = ×400. Scale bar = 50 μm. **g** Paris grade of shed cells before or after alpha1–oleate instillation. An increase is observed in the treatment group ($\chi^2$ test). **h** Reduction in tumor size in patients receiving alpha1–oleate treatment. Images were compared between the time of diagnosis and the time of TURB ($P = 0.04$, $\chi^2$ test for trend compared to placebo, $n = 19$ for treatment group and $n = 20$ for placebo group). **i** Examples of cystoscopy photographs obtained by A.B. at the time of diagnosis and after treatment at the time of TURB. Scale bars = 5 mm. *$P < 0.05$, **$P < 0.01$, ***$P < 0.001$, ****$P < 0.0001$. The data were analyzed by two-tailed unpaired Mann–Whitney $U$-test (**a**, **d**) or by repeated-measures two-way ANOVA with Sidak's correction (**b**, **e**).

Cancer cells are aggressive, outcompete healthy cells, and ruin tissue integrity. It is generally assumed that treatments must be equally aggressive and highly toxic substances are often used, despite their lack of selectivity and the severe side effects that they cause. The protein–lipid complexes studied here are attractive to cancer cells, which actively internalized them, but end up being killed. Healthy cells are less responsive and extensive toxicity studies have failed to detect adverse effects in the bladder[13]. This low toxicity was confirmed here, as no drug-related side effects were observed in the treatment group. The results, therefore, identify the alpha1–oleate treatment of NMIBC as an interesting therapeutic option. In view of the low toxicity observed so far, liberal intra-vesical administration in early-stage NMIBC might be an interesting approach to postponing the introduction of more toxic and invasive therapeutic options.

## Methods

### Clinical trial of alpha1–oleate in NMIBC

*Trial design and patient population.* This was a single-center, placebo-controlled, double-blinded randomized Phase I/II interventional clinical trial of non-muscle invasive bladder cancer taking place from May 21, 2018 to June 3, 2019. Subjects diagnosed with non-muscle invasive bladder cancer and scheduled for transure-thral surgery were included in the study. The study was approved by the State Institute for Drug Control (SUKL) in the Czech Republic; number 273799/17-I and the Ethics Committee of the Motol University Hospital; number EK-786/17 (ClinicalTrials.gov Identifier: NCT03560479). Patients gave their written informed consent.

*Peptide synthesis and alpha1–oleate complex generation.* Peptide synthesis and the preparation of the investigational product were performed under good manu-facturing practice (GMP) conditions and the complex was diluted in phosphate-buffered saline (PBS) to the final concentration (1.7 mM). The placebo group received PBS (sodium chloride, potassium chloride, sodium- and potassium phosphate, and water for injection), which was identical in appearance to the active treatment.

*Study protocol.* Study subjects were randomized 1/1 and received intra-vesical instillations (30 mL) of either alpha1–oleate (1.7 mM) or PBS on six occasions during a period of 22 days (days 1, 3, 5, 8, 15, and 22). A safety follow-up was included 52 days after the last instillation (EudraCT Number: 2016-004269-14 and ClinicalTrials.gov NCT03560479). The complete Study Protocol is provided as Supplementary Note 1 in the Supplementary Information file. The interim analysis of the clinical trial presented here represents a complete evaluation of the Phase I/II study, as per the original protocol. The protocol was later amended to include a 24-month follow-up. At that time, it was decided to perform the initially planned final analysis as an interim analysis, scheduled after all subjects had completed the 52-day safety follow-up. The study underwent data lock and subsequent unblinding was under third-party control. The primary objective of the trial was to evaluate the safety of alpha1–oleate. No formal sample size calculation evaluating the power of the trial has been performed. However, consideration regarding the sample size was made based on a previous open study of HAMLET instillations in bladder cancer patients[16] and in the murine bladder cancer model[14]. For efficacy, the sample size was based on analysis of change in tumor cells assessed before HAMLET

instillation and after 2 h. The mean fold increase of shed cells was 41.3 and the standard deviation was 60.4 in 9 patients[16]. A sample size of 20 patients per group was deemed suitable to achieve criterion for significance (alpha) 0.05 and power 90% using the paired samples 1-tailed $t$-test. The null hypothesis is $H_0$: mean change in cell shedding = 0 and the alternative hypothesis is $H_A$: mean change in cell shedding > 0.

Demographic data, morbidity and health parameters as well as tumor characteristics were recorded by the study physicians in the electronic Case Report Form (eCRF) and closely monitored by an external monitor. Population characteristics were evaluated by the study statistician. No significant differences between the treatment and placebo groups were registered in terms of age, gender, co-morbidity, or tumor parameters.

*Primary endpoints.* - Safety as AEs profile (time frame: from the signing of informed consent (day 1) and until end of study (day 52)): Incidence of AEs and classification in terms of severity, causality, and outcome.

- Efficacy as cell shedding (time frame: days 1–22): change in cell shedding into urine (number of epithelial cells per mL of urine).

- Change from baseline in characteristics of papillary tumors (time Frame: prior to treatment (baseline) and on day 30, in connection with scheduled surgery): the bladder tumors are characterized by in vivo imaging during examination by cystoscopy.

*Secondary endpoints.* - Histopathology scoring of the tumor using established parameters for scoring of Grade and Stage/Invasiveness.

- Urine cytology examined before and after instillation, using the Paris scoring system.

- Uptake of alpha1–oleate by tumor cells, defined by staining with specific antibodies.

- Tissue apoptotic response to alpha1–oleate, defined by TUNEL staining.

- Tumor response to alpha1–oleate, defined by RNA sequence analysis.

- Proteomic analysis of markers in urine was not completed.

- Long-term effects of the study treatment have not been evaluated.

*AEs profile.* AEs were collected from the signing of the informed consent form until the end of the study (FU1 Visit, day 52). All diagnoses, symptom(s), sign(s), or finding(s) with a start date after the first dose of the study drug were recorded as AEs or severe AEs (SAEs). (S)AEs related to the study procedure were coded during the course of the trial according to MedDRA by preferred terms and pri-mary system organ class. All AEs recorded during the course of the trial were included in the subject data listings and an overall summary of the number (percentage) of subjects with any treatment-emergent (S)AEs, premature dis-continuations from the trial due to AEs, treatment-related AEs, and SAEs were constructed. The number of subjects experiencing each type of adverse event was tabulated regardless of the number of times each adverse event was reported by each subject. The severity of each type of adverse event was also tabulated and graded as the most severe recording for that adverse event.

*Cell shedding.* To quantify the shedding of cells and cell clusters into the urine, samples were obtained from each patient prior to and after each instillation of alpha1–oleate or placebo (Visit 1–Visit 6). Cell shedding was quantified by counting the total number of epithelial cells in a unit of uncentrifuged urine under light microscopy, using a hemocytometer chamber. Changes in cell shedding were quantified at each visit, by comparing cell numbers in samples obtained before and after each instillation. The cell clusters were scored based on the examination of these samples by an experienced pathologist on a range of 0–2 where 0 = no clusters and 2 = the highest number of clusters.

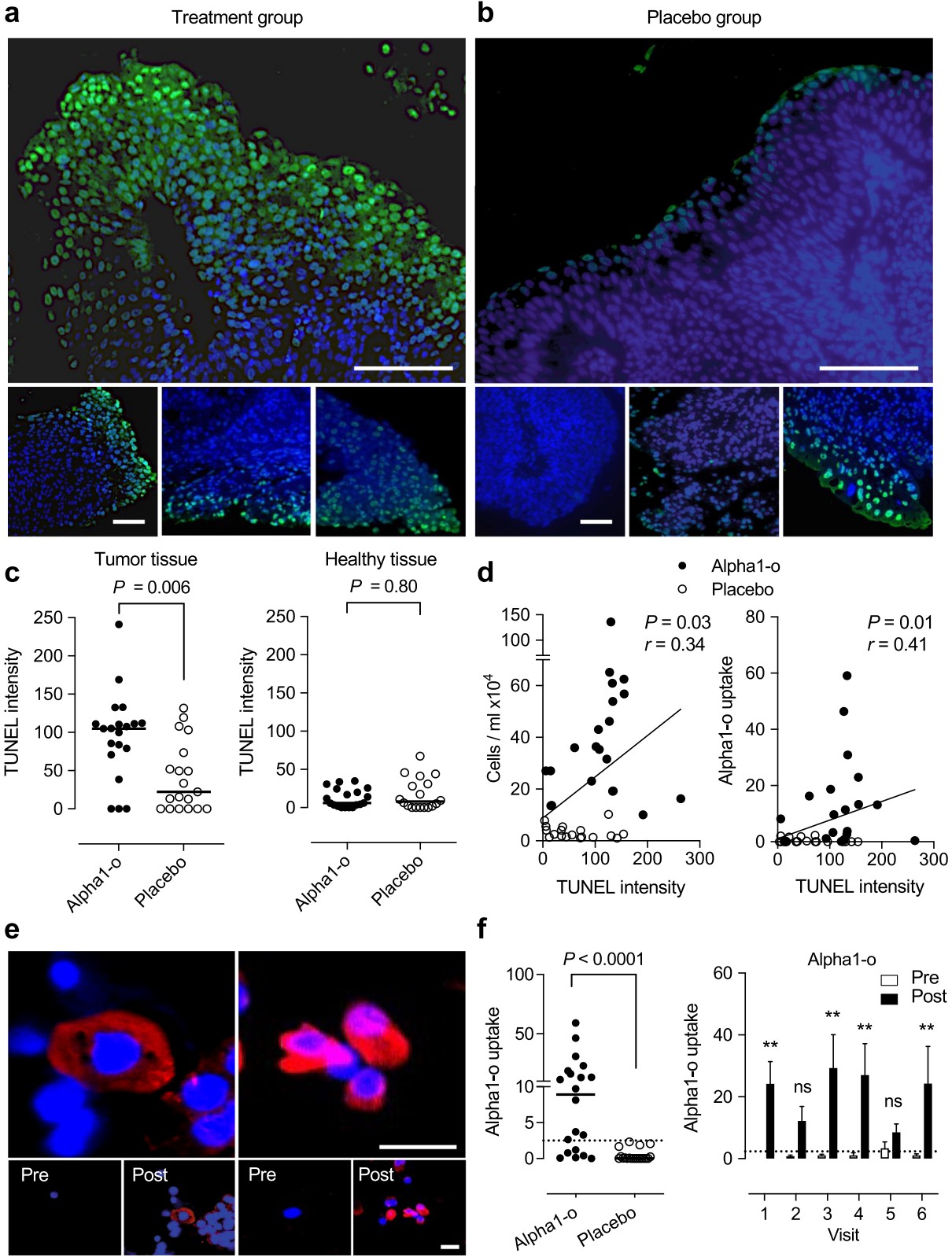

*Characteristics of papillary tumors, tumor size*. To examine if the alpha1–oleate treatment affects tumor size, all included subjects underwent outpatient cystoscopy at Visit 0. Tumors were reexamined at Visit 7, prior to scheduled surgery. High-quality photographs were collected endoscopically, using a flexible cystoscope (Olympus) before being removed by TURB according to EAU Guideline recommendations[42]. Changes in tumor size were evaluated intra-individually, using paired images. The results were evaluated using a simplified Delphi method[23], by an independent NMIBC expert. Changes in lesion size, superficial necrosis, and tissue vascularization were addressed in a blinded manner.

*Histopathology scoring*. Tumor biopsies, collected at the time of surgery were evaluated by histopathology, using established parameters for scoring of Grade and Stage/Invasiveness. Tissue samples were analyzed by a designated study uro-pathologist. Both grading classifications (WHO 1973 and 2004/2016) were used[43].

**Fig. 6 Apoptotic response to alpha1–oleate and cellular uptake by tumor cells.** Apoptosis was quantified in tumor biopsies, using the TUNEL assay. Arbitrary units were calculated after subtraction of background staining in TUNEL negative healthy tissue samples. **a** Representative image of TUNEL staining (green = TUNEL, blue = DAPI) in tumor tissue from individual patients receiving alpha1–oleate instillations. Scale bars = 200 μm. **b** Representative images of TUNEL staining in tumor tissue from individual patients receiving placebo. Scale bars = 200 μm. **c** Scatter plot demonstrating elevated TUNEL staining intensity in tumor biopsies from patients receiving alpha1–oleate instillations compared to placebo. TUNEL staining was not significantly altered in healthy tissue biopsies from patients receiving alpha1–oleate instillations or placebo (n = 40 tumors and 38 healthy biopsies, two data points were further removed due to medical conditions from patients and confirmed by Grubbs's outlier test) (two-tailed unpaired Mann–Whitney U-test). Line represents the median. **d** Correlation of TUNEL staining intensity with cell shedding (P = 0.03, 95% CI 0.0220– 0.6010) and alpha1–oleate uptake (P = 0.01, 95% CI 0.0957–0.6461) (Spearman correlation, two-tailed, approximate P-value, n = 20 for alpha1–o and n = 19 for placebo). **e** Representative images of alpha1–oleate (red) uptake with counter-stained nuclei (blue). Alpha1–oleate uptake by tumor cells was quantified by staining of shed cells in urine with polyclonal anti-alpha1–oleate antibodies. Scale bars = 20 μm. **f** Scatterplots of cellular uptake in individual patients receiving alpha1–oleate. Each dot represents the mean fluorescence intensity of six post-instillation samples per patient treated with alpha1–oleate. Comparison of alpha1–oleate uptake by the cell in urine before (pre = white) and after (post = black) alpha1–oleate inoculation on visits 1-6 (repeated-measures two-way ANOVA with Sidak's correction, P = 0.0049 for visit 1, 0.1913 for visit 2, 0.0067 for visit 3, 0.0025 for visit 4, 0.3807 for visit 5 and 0.0043 for visit 6, n = 20 per group and time point). Line represents the median and bars represent mean ± SEM.

Biopsies from healthy tissue areas distant from the tumor were collected for comparison.

*Urine cytology.* Urine cells were centrifuged onto L-lysine-coated microscope slides (Cytospin 3, Shandon) at 113×g for 5 min, fixed and stored at room temperature until further analyses. Urinary cytology was evaluated using the Paris System for Reporting Urinary Cytology 2016[43,44] and defined as: 1. No diagnosis/unsatisfactory. 2. Negative for high-grade urothelial carcinoma. 3. Atypical urothelial cells present. 4. Suspicious for high-grade urothelial carcinoma. 5. High-grade urothelial carcinoma. 6. Low-grade urothelial neoplasm. 7. Other positive for malignancies and miscellaneous lesions.

*Alpha1–oleate uptake.* Alpha1–oleate uptake by tumor cells was quantified by staining with specific antibodies. Cells on cytospin-slides were washed (Tris-buffer saline TBS, 10 min), permeabilized (0.25% TritonX-100 in TBS, 20 min, room temperature) and blocked (5% normal goat serum in TBS, 1 h, room temperature) before the addition of rabbit polyclonal anti-human alpha-lactalbumin antibodies (1:50 in 5% normal goat serum at 4 °C, overnight, Mybiosource, Cat# MBS175270). Slides were washed (TBS, 2 ×5 min) and stained with Alexa-568-labeled secondary antibody (1:200, 1 h, room temperature, ThermoFisher). The nucleus was counterstained using DRAQ5 (1:1000, 15 min) before a final wash (2 ×5 min in TBS). Slides were mounted (Fluoromount aqueous mounting media), before capturing images by laser scanning confocal microscopy (Carl Zeiss). Fluorescence intensity was quantified by ImageJ and net fluorescence calculated after subtraction of the secondary antibody background.

*Apoptosis in tissue biopsies, TUNEL staining.* DNA fragmentation was detected using the terminal deoxynucleotidyl transferase dUTP nick end-labeling (TUNEL) assay (Click-iT TUNEL Alexa Fluor 488 imaging assay kit, ThermoFisher). Tissue sections were de-paraffinized with xylene followed by serial dehydration with ethanol (100%, 95%, 75%, and 50%). Dehydrated sections were fixed (4% PFA, 15 min), permeabilized (DNase-free Protease K solution 20 μg/mL, 15 min) and incubated with TUNEL reaction mixture containing TdT for 60 min at 37 °C. After the TUNEL reaction, sections were incubated with Click-iT reaction mixture (30 min, 37 °C). Sections were counterstained with DAPI (1 μg/mL, 5 min), mounted in Fluoromount aqueous mounting media, and analyzed by fluorescence microscopy (Zeiss). Fluorescence intensities were quantified by ImageJ and net mean fluorescence intensity calculated after subtraction of background fluorescence.

### RNA sequence analysis of tissue biopsies.

RNA was extracted from tissues stabilized in RNAlater using the AllPrep DNA/RNA/miRNA Universal Kit. Disruption was in the TissueLyser system and CK28 Precellys tubes and by homogenization in the QIAshredder homogenizer. The quantity and quality of the RNA samples were evaluated using NanoDrop and Agilent 2100 Bioanalyzer. RNA samples were prepared by Illumina TruSeq Stranded mRNA Library Prep Kit (20020594), and libraries were multiplexed and sequenced using NextSeq 500/550 High Output Kits (v2.5 2 × 75 Cycles) with an average of 22 million reads per sample. Raw sequencing data were demultiplexed using bcl2fastq (version 2.18) and RSEM (1.3) was used for abundance estimation using the human genome release 37/Ensemble 75. Samples were thoroughly quality checked (QC) and visualized using dimensionality reduction (i.e. PCA), MA-plots as well as RNA-seq intrinsic biases (such as GC bias, transcriptome complexity, and alignment quality). Differential expression analysis was performed using R (version 3.4) and the packages limma and DESeq2. Fold changes were calculated by comparing tumors in the treated to the placebo group. Relative expression levels were analyzed and genes with an absolute fold change >2.0 and P < 0.05 were considered as differentially expressed. Heat-maps were constructed using the Gitools 2.1.1 software.

Differentially expressed genes were functionally characterized using the Ingenuity Pathway Analysis version 57662101 (IPA, Qiagen) software.

### Statistical analysis.

For efficacy, the sample size was based on analysis of change in tumor cells assessed from a previous study[16]. A sample size of 20 patients per group was deemed suitable to achieve criterion for significance (alpha) 0.05 and power 90% using the paired samples 1-tailed t-test. The null hypothesis was $H_0$: mean change in cell shedding = 0 and the alternative hypothesis is $H_A$: mean change in cell shedding > 0. The Gaussian distribution was determined by the D'agostino and Pearson normality test. For data following a Gaussian distribution, student t-tests were used. Other data sets were analyzed by Mann–Whitney U-test. Correlations were determined by Spearman correlation. Kinetic data were analyzed using the repeated measures two-way ANOVA test. All statistical analysis was done by using Prism version 6.02 (GraphPad Software Inc.). P values < 0.05 were considered statistically significant. All images were created by the study team.

### Molecular and cellular studies

*Chemicals and antibodies.* Sodium oleate (Sigma-Aldrich, Cat# O7501), Alexa-Fluor568 protein labeling kit (Thermo Scientific, Cat# A10238), AlexaFluor488 protein labeling kit (Thermo Scientific, Cat# A10235), ATPlite (Perkin Elmer, Cat# 6016947), Presto Blue Cell Viability Assay (Invitrogen, Cat# A13262), Anti-peptide antibodies (this study, produced by GeneCust), FluxOR Potassium ion channel assay (Invitrogen, Cat# F20015), Barium chloride $BaCl_2$ (Sigma-Aldrich, Cat# B0750), Amiloride (Sigma-Aldrich, Cat# A7410), Click-iT TUNEL Alexa Fluor 488 imaging assay kit (ThermoFisher Scientific Cat# C10245), DRAQ5 (Abcam, Cat# ab108410), Fluoromount (Sigma-Aldrich, Cat# F4680), DNA/RNA/miRNA Universal Kit (Qiagen, Cat# 80224),

*Peptide synthesis and complex generation.* Peptides for in vitro and in vivo experiments were synthesized using Fmoc solid-phase chemistry (Mimotopes, Melbourne, Australia). For biotinylated peptides, an aminohexanoic acid (Ahx) spacer was added to ensure adequate separation between the biotin and the peptide moieties. A five-fold stoichiometric concentration of sodium oleate in phosphate-buffered saline was prepared, followed by the addition of each respective peptide. The more hydrophobic peptides were initially dissolved in DMSO, then transferred to the oleate buffer. The sequences for the peptides are as follows:

| | |
|---|---|
| Alpha1: | Ac-KQFTKAELSQLLKDIDGYGGIALPELIATMFHTSGYDTQ-OH |
| Beta: | Ac-IVENNESTEYGLFQISNKLWAKSSQVPQSRNIADISADKFLD DD-OH |
| Sar1alpha: | Ac-MAGWDIFGWFRDVLASLGLWNKH-OH |
| Sar1beta: | Ac-DRLATLQPTWHPTSEELAIGNIKFTTFDLGGHI-OH |

*Cell lines and cell culture.* Human lung carcinoma cells (A549, ATCC Cat# CCL-185, RRID:CVCL_0023), human kidney carcinoma cells (A498, ATCC Cat# HTB-44, RRID:CVCL_1056), and murine bladder carcinoma cells (MB49, RRID: CVCL_7076, provided by Sara Mangsbo, Uppsala University, Sweden) were cultured in RPMI-1640 with non-essential amino acids (1:100), 1 mM sodium pyruvate, 50 μg/mL gentamicin, and 5–10% fetal calf serum (FCS) at 37 °C, 5% $CO_2$.

*Cell death assays.* To quantify effects on cell viability, A549, A498, or MB49 cells were seeded in 96-well plates ($2 \times 10^4$/well, Tecan Group Ltd.), cultured overnight at 37 °C, 5% $CO_2$ and incubated with peptide–oleate complexes in serum-free

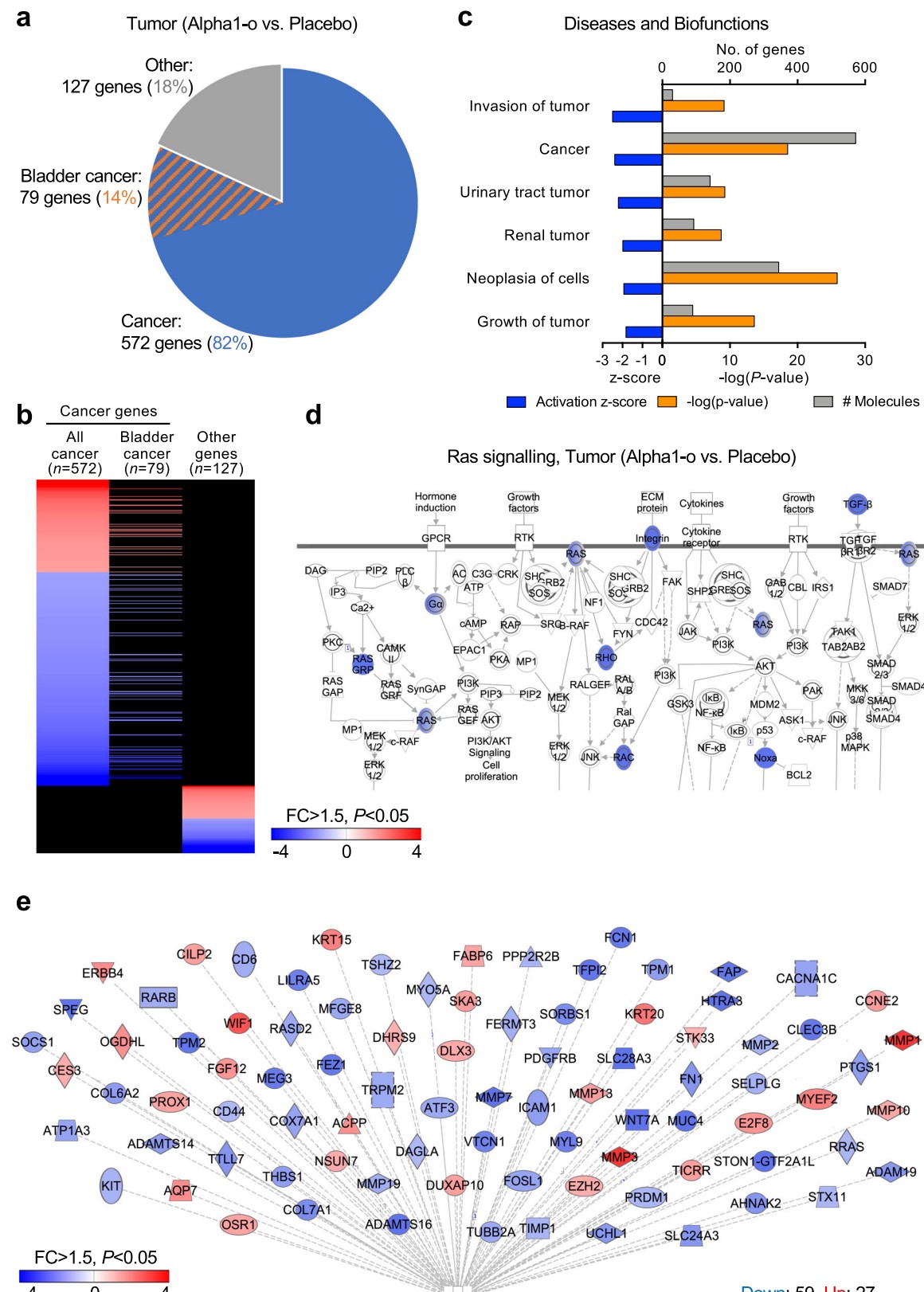

RPMI-1640 at 37 °C. FCS was added after 1 h and cell viability was quantified after 3 h, by the Presto Blue fluorescence assay (Thermo Scientific) or ATP levels (luminescence-based ATPlite kit), using a microplate reader (Infinite F200, Tecan).

The colony assay was used to define long-term effects on cell viability. MB49 cells were seeded in a 24-well tissue culture plate (1000 cells/well in 1 mL in RPMI with FCS), allowed to adhere for 24 h, washed in PBS and treated with alpha1–oleate or sar1alpha–oleate (7, 21, or 35 μM in 1 mL of RPMI without FCS

for 1 h). After the addition of FCS (5%) the cells were incubated at 37 °C in 5% $CO_2$ for 10 days or until visible colonies were present in the PBS control. Plates were fixed in cold methanol, stained with hematoxylin. Staining intensity was quantified using ImageJ.

TUNEL staining was performed using Click-iT TUNEL Alexa Fluor 488 Imaging Assay kit (Thermo Scientific). Briefly, the cells were fixed in 2% paraformaldehyde for 15 min followed by permeabilization with 0.25% TritonX-

**Fig. 7 Reprogramming of gene expression.** RNA sequencing was used to compare gene expression profiles in tumor tissue biopsies from the treatment or placebo groups. **a** Pie chart of genes regulated in response to treatment (cut-off FC > 1.5, $P < 0.05$ compared to the placebo group). In the treatment group, 82% of all regulated genes were cancer-related and 14% were bladder cancer-related. Gene categories were identified by biofunction analysis. **b** Heatmap of specific cancer- and bladder cancer-related genes regulated in tumor biopsies from the treatment group (red = upregulated, blue = downregulated, cut-off FC > 1.5, $P < 0.05$ compared to placebo group). About 60% of all regulated genes were inhibited in the treatment group. **c** Detailed analysis of data in (**a**, **b**). Top regulated, cancer-associated functions are shown. Inhibition is indicated by negative $z$-scores (blue) and significance by $P$ values (orange). The expression of genes involved in tumor invasion, neoplasia, tumor growth, and urinary tract tumors was strongly inhibited. **d** Inhibition of Ras signaling in the treatment group compared to placebo. **e** Bladder cancer gene network regulated specifically in patients receiving alpha1–oleate treatment compared to placebo.

100 for 20 min. The sections were incubated for 60 min at 37 °C with the TUNEL reaction mixture, then counter-stained with DAPI (20 µg/mL) for 15 min. A positive control slide (cells treated with DNase I for 30 min at 37 °C) was included in each experiment. The TUNEL fluorescent intensity was quantified using Image J.

*Membrane blebbing and ion fluxes.* Membrane changes in lung carcinoma cells were visualized by transmission light microscopy imaging. Cells were seeded on a glass coverslip and allowed to partially adhere to the glass surface for 10 min at room temperature, prior to exposure to the alpha1–oleate or sar1alpha–oleate complexes. Changes in cell morphology were captured with LSM 510 META confocal microscope (Carl Zeiss) using ×40 oil immersion objective. Naked peptides and oleate were used as negative controls (Supplementary Fig. 1e).

$K^+$ fluxes were quantified using the FluxOR potassium ion channel assay (Invitrogen). The influx of the indicator $Ti^+$ measures the opening of $K^+$ channels[45]. Briefly, 20,000 adherent cells were incubated in loading buffer followed by incubation with assay buffer and stimulus buffer for 60 min. For inhibition, cells were pretreated with the $K^+$ channel inhibitor $BaCl_2$ (100 µM) followed by treatment with alpha1–oleate (35 µM). Fluorescence was measured at 535 nm after excitation at 485 nm using a fluorescence plate reader (TECAN infinite F200).

### Structural analysis of the peptide–oleate complexes

*CD spectroscopy.* Far-ultraviolet (UV) CD spectra were collected on alpha1-, beta-, sar1alpha-, and sar1beta-peptides with and without oleate at 25 °C using a Jasco 815 CD Spectropolarimeter. The peptides were dissolved in 50 mM sodium phosphate buffer, pH 7.4, with 10% $D_2O$, at a final concentration of 0.2 mg/mL. Far-UV CD was performed from 185 to 260 nm for the samples without oleate and from 200 to 260 nm for the samples with oleate and the background was subtracted. The mean residue ellipticity (MRE), $[\theta]$, in deg $cm^2$/dmol, was calculated as described previously[9].

*Biomolecular NMR spectroscopy.* The alpha1- and sar1alpha-naked peptide samples were dissolved in 50 mM sodium phosphate buffer (pH 7.4, 90% $H_2O$, 10% $D_2O$), and the peptide–oleate complexes were reconstituted from a lyophilisate of phosphate-buffered saline. All experiments were carried out in the phase-sensitive mode[46]. One-dimensional $^1H$, two-dimensional NOESY (Nuclear Overhauser Effect Spectroscopy) and $^1H$–$^{13}C$ HSQC (Heteronuclear Single Quantum Correlation) spectra were acquired on an Agilent Technologies 18.8T (800 MHz) DD2 Premium Compact spectrometer with a triple-resonance, 5 mm enhanced cold probe. The $^1H$–$^{13}C$ HSQC spectra were collected at 20 °C with 16 scans, an initial delay of 3.0 s, a 90° pulse width of 7.5 and 9.8 µs, and an acquisition time of 0.4 s with broadband decoupling for alpha1 and sar1alpha peptide samples. For the alpha1–oleate and sar1alpha–oleate complexes, we used a 90° pulse width of 12.80 and 13.30 µs and an acquisition time of 0.4 s. All acquisition parameters were kept constant for all samples. Two-dimensional DPFGSE-NOESY (Double Pulse Field Gradient Spin Echo-NOESY) pulse sequences were used to acquire data at 20 °C with 16 scans, with an optimized mixing time of 300 ms for the alpha1 and alpha1–oleate complexes and a delay period of 1.5 s. For the sar1alpha–oleate complex, water-gate NOESY was used with 12 scans, with a mixing time of 150 ms. A trace amount of TSP was added to serve as a chemical shift reference. Each 2D HSQC spectrum consisted of 4 K complex points in the acquisition dimension and 512 complex points in the indirect dimension. For the NOESY spectra, 4 K complex points were used in the acquisition dimension and 1 K complex points in the indirect dimension. The two-dimensional data were processed with Gaussian apodization in both dimensions. The stoichiometry of the peptide with the oleic acid was determined by comparing the peak areas (using the 1D 1H spectra) or peak volumes (using the 2D $^1H$–$^{13}C$ HSQC spectra) of well-resolved, isolated regions found in the spectra.

Diffusion-ordered spectroscopy (DOSY) measurements were performed at 293 K. Samples were prepared in 50 mM phosphate buffer at pH 7.4. The DgscteSL_dpfgsc DOSY pulse program was used, which consists of gradient compensated stimulated echo with spin lock using the excitation sculpting solvent suppression method[47]. A spectral window of 13,020 Hz was used, with an acquisition time of 2.46 s with a relaxation delay of 3 s. The FIDs were collected with 32,000 complex data points with 64 scans. Logarithmically the gradient pulse strength was increased from 3% to 86% of the maximum strength of 32,767 G/cm

in 60 steps. A diffusion time ($\Delta$) of 100 ms and bipolar half-sine-shaped gradient pulses ($\delta$) of 5 ms was applied. 1,4-Dioxane, which is known to behave independently of protein concentration and the folded state of the protein, was used as an internal chemical shift reference and hydrodynamic radius calibration reference (3.75 ppm; $R_H$ = 2.12 Å)[48,49]. DOSY processing was performed using a two-component fit with a discrete approach, which further processed using a non-uniform gradients approach. Three replicate acquisitions were given for each sample, and the resulting diffusion coefficient ($D$) values calculated. For alpha1 peptide the average $D$ value was 2.162 and 14.10 $m^2$/s for 1,4-dioxane. In the case of alpha1–oleate complex the average $D$ value was 0.986 $m^2$/s for complex and 13.61 $m^2$/s for 1.4-dioxane. The calculated $R_H$ are as follows: alpha1 peptide $R_H$ = 13.82 ± 0.447 Å, alpha1–oleate complex $R_H$ = 29.3 ± 0.606 Å, HSA $R_H$ = 40.9 ± 1.44 Å, oleate in aqueous solution $R_H$ = 104.3 ± 7.22 Å, oleate in methanol $R_H$ = 5.58 ± 0.0649 Å. Note that the $D$ (diffusion coefficient) values for 1,4-dioxane are slightly variable dependent upon the co-solute (lower panel where $D$ is between 14.4 and 12.6), which rightly reflects the different solution micro-environment conditions that both solutes are mutually experiencing for each sample.

For $T_2$ relaxation measurements, the standard CPMGT2 pulse sequence was used to run the experiments with 15 relaxations delays, which were chosen logarithmically for different maximum $T_2$ time intervals: 8 s (alpha1 peptide), 1.2 s (alpha1–oleate complex), 3.0 s (HSA), 7.0 s (oleate in aqueous solution), and 10 s (oleate in methanol), respectively. The data were acquired with 32,000 complex points with a baseline correction of 4. The $T_2$ analyses were performed on VNMRJ version 4.0 (Agilent Technologies) software by the exponential fitting of these values with their corresponding intensity. All other NMR parameters were kept constant for all samples throughout the experiments. The experiments were acquired at a sample temperature of 293 K. The data are presented in Supplementary Table 1.

*Size-exclusion HPLC.* Calibration standards and samples were injected onto a TSKgel Super SW3000 HPLC column (4.6 mm × 30 cm, Particle size 4 µm, pore size 25 nm, Tosoh Bioscience) eluted with 0.05 M sodium phosphate buffer pH 7.0 containing 0.1 M $Na_2SO_4$ at a flow rate of 0.25 mL/min and detection at 280 nm. The chromatography was performed on a Dionex Ultimate HPLC 3000 Standard System running Chromeleon 6 software (Dionex, Thermo Scientific). The standard calibration curve was generated with the proteins given including HSA, which has a hydrodynamic radius ($R_H$) of 40 Å (Supplementary Fig. 5). The $R_H$ vs. elution volume linearity of the standard calibration curve is known to vanish after approximately 3.8 mL of elution volume[50]. As a result, the $R_H$ of oleate in methanol eluent (a solvent that ensures that the fatty acid is monomeric) will be less than what is estimated from the standard curve (12.3 Å). The retention times of small $R_H$-analytes are closely reproduced regardless of eluent, be it aqueous buffer or methanol.

*Computational simulations: model building of peptide and peptide–OA complexes.* The initial structure of the alpha1 peptide was obtained from the corresponding domain in the crystal structure of human alpha-lactalbumin (PDB ID: 1B9O). All cysteines were mutated to alanines, consistent with findings that a reduced human alpha-lactalbumin mutant in which all cysteines mutated to alanines could form a cytotoxic complex in the presence of the lipid cofactor[9]. The initial structure of the sar1alpha peptide was obtained from an I-TASSER-built homology model[51]. The sequence similarity of the Sar1alpha peptide with the sequence of the top-ranked threading template used by I-Tasser (PDB ID: 1R7G) was 0.35. However, this is eventually irrelevant as we used extensive H-REMD sampling to sample the conformations of the peptide, both in its oleate bound and apo forms. The alpha1 and the sar1alpha peptide were centered in a cubic box with box edges 1.2 nm from the peptide. For the oleate-containing systems of alpha1 and sar1alpha, 4 molecules of oleate are placed randomly in the box surrounding the alpha1 peptide to obtain a peptide–oleic acid ratio of 1:4. The Amber 99SB-ildn[52] force field and the TIP3P[53] water model were used. For the coordinates, the starting structure was built using Discovery Studio 4.1 (Accelrys). Geometry optimization for the ligand was performed using Gaussian09[54] at the level of HF-6-31G*, and the partial charges were determined by the RESP[55] method implemented in the antechamber tool of AmberTools16 (AMBER 2016). Topologies for the oleate were built using the General Amber Force Field[56]. The respective atom labels, corresponding atom type, and partial charges are shown in Supplementary Table 10. Additional parameters

for the molecule are derived from the General Amber Force Field. For reference, the structure of the oleate molecule and its atom labels is shown in Supplementary Fig. 11. All systems were neutralized and Na$^+$ and Cl$^-$ ions were added to a concentration of 0.15 M. Energy minimization was performed using the steepest descent algorithm for 1000 steps to remove any initial bad contacts. Long-range electrostatics were treated with the particle mesh Ewald algorithm[57], with a real-space cutoff of 1.2 nm, and Van der Waals interactions were truncated at 1.2 nm. All systems with oleate-containing peptides or naked peptides were initially heated at 500 K for 40 ns to eliminate starting structure bias and provide a partially unfolded state for the peptides. Temperature coupling of the system was performed using a velocity rescaling thermostat[58].

*Hamiltonian replica exchange molecular dynamics simulations.* The Gromacs 5.1.2 molecular dynamics package[56] with the Plumed 2.3 plugin for Hamiltonian Replica Exchange Molecular Dynamics[59] was used to perform the simulations. All atoms of the alpha1- and sar1alpha peptide residues, along with the oleate residues of the two oleate-containing systems, were selected for Hamiltonian scaling. Twenty replicas were used for each system, and scaling factors were generated for an effective temperature range of 300–800 K. Temperatures for scaling were selected based on a geometric progression. The temperature factors were 300, 315.893, 332.629, 350.251, 368.807, 388.346, 408.919, 430.583, 453.395, 477.415, 502.707, 529.34, 557.384, 586.913, 618.006, 650.747, 685.223, 721.525, 759.75, and 800 K. Each replica was simulated for 400 ns, resulting in an effective simulation of 8 μs. Exchanges were attempted every 2 ps, and the result was an average acceptance probability of approximately 30%.

*Simulation analysis.* Analysis of simulation data was performed using the built-in Gromacs tools of the Gromacs package[56]. The ensemble for each system with the canonical unscaled potential energy was used for the analysis, and data analysis was performed on the last 300 ns for each system. Dihedral principal component analysis[60] was performed using the gmx angles, gmx covar, and gmx anaeig tools to prepare and diagonalize the covariance matrix and analyze eigenvectors and eigenvalues. The free-energy surface was constructed through projection onto the first and second principal components with the formula $F_i = -RT \ln(P_i/P_o)$, where $R$ is the gas constant, $T$ is temperature (300 K), $P_i$ is the population in each bin and $P_o$ is the population of the most populated bin. The gmx cluster tool of Gromacs 5.1.2 was used to identify the representative structure of each minima for geometric clustering for the Gromos algorithm. We used the define secondary structure of proteins (DSSP)[61] algorithm to calculate secondary structure propensities. For our analysis, we classified the 3$_{10}$ helix, α helix, and π helix structures as helices; the β-sheet and residue in isolated β-bridge structures as sheets and the remaining structures as others. The contact probability was calculated using the gmx mindist tool in the Gromacs package. The minimum distance between protons of side chains for each residue and oleic acid was calculated for each frame. To calculate the contact probability, a contact was defined if the measured distance was less than 0.55 nm. The contact probabilities between Aromatic ring protons and Olefinic protons of alpha1- and sar1alpha–oleate-containing systems were also calculated similarly. Proton distances were calculated to facilitate the comparison of simulation data to Nuclear Overhauser Spectroscopy data.

**Reporting summary.** Further information on research design is available in the Nature Research Reporting Summary linked to this article.

## Data availability
The data supporting the structural and cellular findings of this study are available within the article and its supplementary information files. The structural data referenced during the study are available in a public repository from the Protein Data Bank website (www.rcsb.org, DOI:10.2210/pdb1B9O/pdb, DOI:10.2210/pdb1R7G/pdb). The RNA sequencing data generated in this study have been deposited in the Gene Expression Omnibus (GEO) database under accession number GSE172112. Source data are provided with this paper.

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

## Acknowledgements

The authors thank Petr Bouška and NEOX for the Monitoring of the clinical study; Jeanette Valcich and the Center for Translational Genomics (CTG) at Lund University for the mRNA Library prep and Sequencing; Susanne Strömblad, Lina Gefors and the Lund University Bioimaging Centre (LBIC) for providing experimental resources for tissue analysis, Arunima Chaudhuri for input and comments on the manuscript, Sara Mangsbo, Uppsala University, Sweden for the MB49 cells (RRID:CVCL_7076). We gratefully acknowledge the support of the Swedish Research Council, the Swedish Cancer Society (Cancerfonden) and HAMLET Pharma. Support to the Svanborg group was further provided from the European Union's Horizon 2020 research and innovation program under grant agreement No. 954360. The funding sources had no role in the design of this study or in its execution, analyses, interpretation of the data, or decision to submit the results for publication.

## Author contributions

All authors met the ICMJE criteria for authorship. C.S., M.B., J. Ho, Y.G.M., K.H.M., conceived and designed the study. J. Ho, P.S.K., A.H., D.L.F., K.H.M. performed structural biology experiments and analyses. J.T.Y.-N. performed and analyzed the simulations under Y.G.M. supervision. J. Ho, A.N., T. Hiep T., P.E. performed cellular experiments and analyses. A.B., J. Háček, I.A., D.S.C.B., M.L.Y.W., T. Hiep T., H.N., J. Horňák, M.B., and C.S. performed the human trial and analyses. M.B., A.B., J. Háček, I.A., D.S.C.B., T. Hiep T., T. Hien T., M.L.Y.W., P.S., M.B., C.S. analyzed the data and J. Ho, I.A., D.S.C.B., T. Hiep T., K.H.M., M.L.Y.W., A.B., M.B., C.S. wrote the paper.

## Funding

## Competing interests

C.S. holds shares in HAMLET Pharma, as a representative of scientists in the HAMLET group. Patents protecting the use of the alpha1 peptide were filed previously (Biologically active complexes and therapeutic uses thereof; GB 201707715 priority date 14/05/2017, PCT/EP2018/062396 filing date 14/05/2018; inventors: C.S., A.N., J.Ho). No specific patents have been filed based on this study. Other authors declare no competing or conflicts of interests.
