## [Peer Review File · Nature Communications]

Reviewers' comments:

Reviewer #1 (Remarks to the Author); expert on urothelial cancer clinics:

This is an interesting paper describing the potential role of protein-lipid complexes as an intravesical therapeutic option in NMIBC. The clinical aspects of the study have been well conducted and the results are interesting and deserve further investigation.

I would suggest some further mention in the discussion about a number of other new intravesical agents that have been recently described such as oncolytic viruses and their potential drawbacks when compared to this therapy

Reviewer #2 (Remarks to the Author); expert on molecular simulation:

Overall, the study seems very impressive, but is largely outside my expertise. I restrict my comments to the molecular simulation part, which adds value to the study, but which is a relatively minor part. The simulations have been conducted rigorously, using state of the art of approaches. The analysis is succinctly presented and provides some useful insight, complementing the NMR measurements.

Many aspects of the simulation methodology could have been referenced, but these are all well known in the community.

For the homology model built with I-TASSER, what was the level of sequence similarity with the template sequence?

Force field parameters for the ligand, including (but not limited to) the partial charges that are mentioned should be provided in the Supplementary Information.

In the author contributions, it might be appropriate to note explicitly who performed and analysed the simulations.

Reviewer #4 (Remarks to the Author); expert on cell membrane biochemistry:

This report describes an extensive study of a pair of peptide-fatty acid complexes as anti-cancer agents. The work spans basic biophysical and structural studies, simulation, animal studies, and a clinical trial. Given the broadly interdisciplinary nature of the work, I will limit my comments to those areas in which I have expertise.

While the results are on balance convincing, the breadth of the report makes it difficult to evaluate particular aspects of the science. It might be better to break reports of this work into more tractable quanta. As presented, however, the following areas of concern should be addressed prior to publication:

1. Of most concern is the apparent lack of the proper controls. In all experiments, the only control performed against the peptide-oleate complexes was a blank sample containing neither peptide nor sodium oleate. While this might be understandable in the clinical study, it seems necessary for the biophysical and cellular studies to include controls consisting of bare peptide and sodium oleate without peptide. This is especially the case in the GUV and membrane blebbing studies.
2. The claim of these complexes as "targeted" therapeutics seems overstated. While they may have a favorable toxicity profile, there is no designed or investigated mechanism of targeting. In fact, the computational studies show the "complexes may potentially be interacting with multiple putative binding partners."

Point-by-point response to the reviewers

Reviewer #1 (Remarks to the Author); expert on urothelial cancer clinics:

This is an interesting paper describing the potential role of protein-lipid complexes as an intravesical therapeutic option in NMIBC. The clinical aspects of the study have been well conducted and the results are interesting and deserve further investigation.

I would suggest some further mention in the discussion about a number of other new intravesical agents that have been recently described such as oncolytic viruses and their potential drawbacks when compared to this therapy

We thank the reviewer for these positive comments. A paragraph about new intravesical therapies such as hyperthermia or oncolytic viruses has been added to the Discussion and referenced. The potential strengths and drawbacks when compared to this therapy have been discussed. See paragraph below.

'Novel therapeutic tools are actively being developed and tested in patients with NMIBC, particularly in patients with disease recurrence after BCG treatment³⁵⁻³⁷. Device assisted hyperthermia was shown to increase the efficacy of intra-vesical chemotherapy but treatment was accompanied by side effects, reducing compliance³⁸⁻⁴⁰. A novel oncolytic-virus-based intra-vesical therapy was reported to achieve a complete response in 53.4% of patients with BCG-unresponsive carcinoma in situ, in a phase III trial⁴¹. The authors discuss the assessment of side effects and the development of biomarkers to help select patients suitable for this therapy. In patients with BCG unresponsive disease, treated with systemic Pembrolizumab, a 41% response rate was reported but side effects were prevalent, limiting compliance (Keynote-676 trial³⁵). The present study identifies alpha1-oleate as an active drug candidate with low toxicity. Further dose-finding clinical studies and adjuvant therapy protocols will be essential to define the therapeutic window of this complex.

Cancer cells are aggressive, outcompete healthy cells and ruin tissue integrity. It is generally assumed that treatments must be equally aggressive and highly toxic substances are often used, despite their lack of selectivity and the severe side effects that they cause. The protein-lipid complexes studied here are attractive to cancer cells, which actively internalized them, but end up being killed by an apoptosis-like mechanism. Healthy cells are less responsive and extensive toxicity studies have failed to detect adverse effects¹³. This low toxicity was confirmed here, as no drug-related side effects were observed in the treatment group. The results therefore identify alpha1-oleate treatment of non-muscle invasive bladder cancer as an interesting new therapeutic option. In view of the low toxicity observed so far¹³, liberal intra-vesical administration in early stage NMIBC might be an interesting approach to postponing the introduction of more toxic and invasive therapeutic options."

References (the numbering follows the main manuscript format):

13. Hien, T. T. et al. Bladder cancer therapy without toxicity—A dose-escalation study of alpha1-oleate. *International Journal of Cancer* 147, 2479–2492, doi:10.1002/ijc.33019 (2020).
35. Kamat, A. M. et al. KEYNOTE-676: Phase III study of BCG and pembrolizumab for persistent/recurrent high-risk NMIBC. *Future Oncology* 16, 507-516, doi:10.2217/fon-2019-0817 (2020).
36. Lamm, D. L. Intravesical Therapy for Superficial Bladder Cancer: Slow but Steady Progress. *Journal of Clinical Oncology* 21, 4259-4260, doi:10.1200/JCO.2003.08.099 (2003).
37. Kamat, A. M. et al. Evidence-based Assessment of Current and Emerging Bladder-sparing Therapies for Non–muscle-invasive Bladder Cancer After Bacillus Calmette-Guerin Therapy: A Systematic Review and Meta-analysis. *European Urology Oncology* 3, 318-340, doi:10.1016/j.euo.2020.02.006 (2020).
38. de Jong, J. J., Hendricksen, K., Rosier, M., Mostafid, H. & Boormans, J. L. Hyperthermic Intravesical Chemotherapy for BCG Unresponsive Non-Muscle Invasive Bladder Cancer Patients. *Bladder Cancer* 4, 395-401, doi:10.3233/BLC-180191 (2018).
39. Colombo, R. et al. Multicentric Study Comparing Intravesical Chemotherapy Alone and With Local Microwave Hyperthermia for Prophylaxis of Recurrence of Superficial Transitional Cell Carcinoma. *Journal of Clinical Oncology* 21, 4270-4276, doi:10.1200/JCO.2003.01.089 (2003).
40. Arends, T. J. H. et al. Results of a Randomised Controlled Trial Comparing Intravesical Chemohyperthermia with Mitomycin C Versus Bacillus Calmette-Guérin for Adjuvant Treatment of Patients with Intermediate- and High-risk Non-Muscle-invasive Bladder Cancer. *European Urology* 69, 1046-1052, doi:10.1016/j.eururo.2016.01.006 (2016).
41. Mueller, T. (Ferring Pharmaceuticals, 2019).

Reviewer #2 (Remarks to the Author); expert on molecular simulation:

Overall, the study seems very impressive, but is largely outside my expertise. I restrict my comments to the molecular simulation part, which adds value to the study, but which is a relatively minor part. The simulations have been conducted rigorously, using state of the art of approaches. The analysis is succinctly presented and provides some useful insight, complementing the NMR measurements.

We thank the reviewer for these positive comments.

Many aspects of the simulation methodology could have been referenced, but these are all well known in the community.

We have included references for the following aspects of the simulation methodology:

- Hamiltonian Replica Exchange Molecular Dynamics: Wang, L., R.A. Friesner, and B.J. Berne, *Replica exchange with solute scaling: a more efficient version of replica exchange with solute tempering (REST2)*. J Phys Chem B, 2011. **115**(30): p. 9431-8.
- Dihedral Principal Component Analysis: Mu, Y., P.H. Nguyen, and G. Stock, *Energy landscape of a small peptide revealed by dihedral angle principal component analysis*. Proteins, 2005. **58**(1): p. 45-52.
- Amber99 SB-ILDN force field: Lindorff-Larsen, K., et al., *Improved side-chain torsion potentials for the Amber ff99SB protein force field*. Proteins, 2010. **78**(8): p. 1950-8.
- TIP3P model: Jorgensen, W.L., et al., *Comparison of simple potential functions for simulating liquid water*. The Journal of chemical physics, 1983. **79**(2): p. 926-935.
- Gaussian 09 software: Frisch, M.J., et al., *Gaussian 09*. 2009, Gaussian, Inc: Wallingford, CT, USA.
- RESP method: Bayly, C.I., et al., *A well-behaved electrostatic potential based method using charge restraints for deriving atomic charges: the RESP model*. The Journal of Physical Chemistry, 1993. **97**(40): p. 10269-10280.
- PME algorithm: Darden, T., D. York, and L. Pedersen, *Particle mesh Ewald: An $N \cdot \log(N)$ method for Ewald sums in large systems*. The Journal of Chemical Physics, 1993. **98**(12): p. 10089-10092.

- V-Rescale thermostat: Bussi, G., D. Donadio, and M. Parrinello, *Canonical sampling through velocity rescaling*. J Chem Phys, 2007. **126**(1): p. 014101.
- DSSP algorithm: Kabsch, W. and C. Sander, *Dictionary of protein secondary structure: pattern recognition of hydrogen-bonded and geometrical features*. Biopolymers, 1983. **22**(12): p. 2577–637.

For the homology model built with I-TASSER, what was the level of sequence similarity with the template sequence?

The sequence similarity of the Sar1alpha peptide with the sequence of the top ranked threading template used by I-Tasser (PDB ID:1R7G) was 0.35. However, this is eventually irrelevant as we used extensive H-REMD sampling to sample the conformations of the sar1alpha peptide, both in its oleate bound and apo forms. Also, as part of our simulation sampling, all systems with oleate-containing peptides or naked peptides were initially heated at 500 K for 40 ns to eliminate starting structure bias and provide a partially unfolded state for the peptides.

This information has been inserted into page 32 of the revised manuscript (methods).

Force field parameters for the ligand, including (but not limited to) the partial charges that are mentioned should be provided in the Supplementary Information. In the author contributions, it might be appropriate to note explicitly who performed and analyzed the simulations.

Geometry optimization for the oleate molecule was performed using Gaussian09 at the level of HF-6-31G*, and the partial charges were determined by the RESP method implemented in the antechamber tool of AmberTools16. Topologies for the oleate were built using the General Amber Force Field. A supplementary table (**Supplementary Table 10**) has been included to show the respective atom labels, corresponding atom type, and partial charges. Additional parameters for the molecule are derived from the General Amber Force Field. For reference, the structure of the oleate molecule and its atom labels has also been included as a supplementary figure (**Supplementary Fig. 11**).

This information has been inserted into page 32 of the manuscript (methods).

In the author contributions, it might be appropriate to note explicitly who performed and analysed the simulations.

The simulations were performed and analyzed by Justin T-Y Ng under the supervision of A/Prof Yuguang Mu. The author contributions were revised to include this information.

Supplementary Table 10. Atom labels, corresponding atom types and partial charges for Oleate molecule.

Atom	Atom Type	Partial Charge	Mass
C1	c2	-0.27428	12
H1	ha	0.13725	1
C2	c2	-0.31519	12
H2	ha	0.13628	1
C3	c3	0.16858	12
H3	hc	0.00472	1
H4	hc	0.00472	1
C4	c3	-0.03521	12
H5	hc	0.00887	1
H6	hc	0.00887	1
C5	c3	-0.01429	12
H7	hc	-0.01022	1
H8	hc	-0.01022	1
C6	c3	0.11368	12
H9	hc	-0.0319	1
H10	hc	-0.0319	1
C7	c3	-0.02086	12
H11	hc	-0.00536	1
H12	hc	-0.00536	1
C8	c3	-0.00953	12
H13	hc	-0.00306	1
H14	hc	-0.00306	1
C9	c3	0.16132	12
H15	hc	-0.02877	1
H16	hc	-0.02877	1
C10	c3	-0.2317	12
H17	hc	0.04852	1
H18	hc	0.04852	1
H19	hc	0.04852	1
C11	c3	0.14538	12
H20	hc	0.00707	1
H21	hc	0.00707	1
C12	c3	-0.0165	12
H22	hc	0.00624	1
H23	hc	0.00624	1
C13	c3	0.00803	12
H24	hc	-0.02199	1
H25	hc	-0.02199	1
C14	c3	0.05504	12

H26	hc	-0.02545	1
H27	hc	-0.02545	1
C15	c3	-0.01683	12
H28	hc	-0.01424	1
H29	hc	-0.01424	1
C16	c3	0.11605	12
H30	hc	-0.0235	1
H31	hc	-0.0235	1
C17	c3	-0.15332	12
H32	hc	-0.01286	1
H33	hc	-0.01286	1
C18	c	0.89604	12
O1	o	-0.8473	16
O2	o	-0.8473	16

Supplementary Figure 11. Structure of oleate molecule with atom labels. Carbon atoms are shown as grey sticks, hydrogen atoms as white sticks, and oxygen atoms as red sticks.

Reviewer #4 (Remarks to the Author); expert on cell membrane biochemistry:

This report describes an extensive study of a pair of peptide-fatty acid complexes as anticancer agents. The work spans basic biophysical and structural studies, simulation, animal studies, and a clinical trial. Given the broadly interdisciplinary nature of the work, I will limit my comments to those areas in which I have expertise.

While the results are on balance convincing, the breadth of the report makes it difficult to evaluate particular aspects of the science. It might be better to break reports of this work into more tractable quanta. As presented, however, the following areas of concern should be addressed prior to publication:

1. Of most concern is the apparent lack of the proper controls. In all experiments, the only control performed against the peptide-oleate complexes was a blank sample containing neither peptide nor sodium oleate. While this might be understandable in the clinical study, it seems necessary for the biophysical and cellular studies to include controls consisting of bare peptide and sodium oleate without peptide. This is especially the case in the GUV and membrane blebbing studies.

In response to the general concerns of the reviewer, **Fig 1** and **Supplementary Fig. 2** have been revised. The new Figures now highlight the efficacy of the synthetic peptide complex in cellular cancer models, using parameters directly relevant for clinical study outcomes, such as cell death and apoptosis.

We thank the reviewer for pointing out the issue of controls and have revised the text to clarify this point.

1. A full set of controls for the cell death assays was included in the submitted manuscript. The beta-oleate complex was used as a negative control in **Fig. 1c,d** and the bare peptides (alpha1 and sar1alpha) and oleate without peptide in **Supplementary Fig. 1**. The effects of oleate have also been separately examined in previous studies (doi:10.1093/molbev/msaa138; 10.1038/srep1643210.1074/jbc.M113.468405). The reported lack of tumoricidal activity of the beta-oleate complex reported here is consistent with these studies (doi:10.1016/j.jmb.2019.05.007; 10.1371/journal.pone.0053051)

2. The naked alpha-lactalbumin peptides and oleate alone were recently shown not to induce large-scale membrane shape changes in the GUV model (doi:10.1093/molbev/msaa138). The GUV data set was therefore removed and the results referenced in the text.

3. Sar1alpha-oleate is shown to induce tumor cell death. The naked Sar1alpha and Sar1beta peptides, in contrast, did not show tumoricidal activity (**Supplementary Fig. 1D**).

4. Alpha1-oleate and Sar1alpha-oleate triggered membrane blebbing in tumor cells (**Fig. 1f** and **Supplementary Figure 2f**). Control experiments have been added, showing that the naked alpha1 and Sar1alpha peptides, the beta-oleate complex and oleic acid alone do not have membrane effects comparable to the complexes (**Supplementary Fig 2**).

2. The claim of these complexes as "targeted" therapeutics seems overstated. While they may have a favorable toxicity profile, there is no designed or investigated mechanism of targeting. In fact, the computational studies show the "complexes may potentially be interacting with multiple putative binding partners."

We apologize if the word "targeted" was used out of context. "Targeted" therapy was mentioned in the introductory paragraph where we argue that many existing "targeted" therapies still cause serious side effects and that increased tumor selectivity is needed. While our data suggest a promising combination of therapeutic efficacy and low toxicity, we have refrained from categorizing alpha1-oleate therapy as "targeted".

Reviewers' comments:

Reviewer #2 (Remarks to the Author):

I raised only minor issues on the original manuscript, which have been satisfactorily addressed in the revised version.

Reviewer #3 (Remarks to the Author):

We congratulate the authors for being responsive to the reviewers' comments. There are only a few minor comments that we recommend the co-authors to restate:

1. In discussion, the authors stated that "... immunotherapy are often suboptimal due to significant side effects and limited efficacy ..." depending if the authors are referring to BCG or immune checkpoint blockade therapy, ICBT are considered relatively efficacious in bladder cancer compared to other cancer types e.g., prostate. This sentence might be overstated, and please revise.

2. In the discussion, the authors might consider discussing the indirect effect of their novel agent in inducing immunomodulatory effects as a potential mechanism, although it is outside the scope of current study e.g., immunogenic cell death etc and include recent citations, and they can stain for CD8+ T cells, dendritic cells and others as a follow up study?

Reviewer #4 (Remarks to the Author):

The revisions to the manuscript adequately address my concerns.

Reviewer #5 (Remarks to the Author):

The manuscript is very well presented and the study conducted covered a broad range of biophysical characterisation, molecular dynamic simulation, cell assays and clinical work on the investigational drug: peptide-lipid complex. The authors have rightly noted its limited use in the presence of lipid binding proteins and hence the use in local administration in this case intravesical administration in NMIBC.

I am happy to recommend the manuscript to be accepted as it is.

Reviewer #6 (Remarks to the Author):

Review of "A novel, peptide-based approach for targeting and killing tumor cells with greater precision"

This work presents a comprehensive set of outcome data from the clinical trial "A First-in-Human Study of alpha1H in Patients With Non-muscle Invasive Bladder Cancer" with three primary endpoints: 1) Incidence of adverse events and classification in terms of severity, causality and outcome; 2) Change in cell shedding into urine and 3) Bladder tumors will be characterized by in vivo imaging during examination by cystoscopy.

My comments are as follows:

- 1) Given that this study was designed as a randomized phase II trial a sample size justification of n=20 per arm should be presented in terms of the statistically detectable effect sizes for each of the three primary aims, i.e. no sample size rational was presented.
- 2) A statistical comparison between Alpha1-o and Placebo should be carried out given the safety

profiles were a primary endpoint. Currently only the descriptive counts are provided in Figure 4c.

3) There were several analyses where there were the factors of treatment and time. The proposed statistical analysis appears to be a two-way ANOVA when in fact it should be a repeated measures ANOVA to account for within subject variation, i.e. a factor for time, treatment and a treatment by time interaction term accounting for within subject correlation.

4) There appears to be missing data in several cases, eg the analysis corresponding to Figure 5i with the tumor size score and the analysis corresponding to Figure 6c. An explanation of missing data should be provided.

5) Are both treatment and control subject data contained in Figure 6d. If so might it be worthwhile to highlight the treatment arm, particularly if there are differential correlations.

6) What were the statistical tests used for figure 1h

7) The statistical plan mentions student's t-tests and the Wilcoxon signed rank test. However, neither of these methods were utilized and hence are not relevant to the current manuscript.

REVIEWER COMMENTS

Reviewer #2 (Remarks to the Author): I raised only minor issues on the original manuscript, which have been satisfactorily addressed in the revised version.

Reviewer #3 (Remarks to the Author): We congratulate the authors for being responsive to the reviewers' comments. There are only a few minor comments that we recommend the co-authors to restate:

1. In discussion, the authors stated that "... immunotherapy are often suboptimal due to significant side effects and limited efficacy ..." depending if the authors are referring to BCG or immune checkpoint blockade therapy, ICBT are considered relatively efficacious in bladder cancer compared to other cancer types e.g., prostate. This sentence might be overstated, and please revise.

This statement in the Discussion was clearly not optimal and has been revised by Professor Babjuk.

"Intravesical chemotherapy and BCG immunotherapy have limited efficacy and significant side effects. Systemic administration of PD-1 and PD-L1 inhibitors is considered only in BCG unresponsive patients and the experience is limited."

2. In the discussion, the authors might consider discussing the indirect effect of their novel agent in inducing immunomodulatory effects as a potential mechanism, although it is outside the scope of current study e.g., immunogenic cell death etc and include recent citations, and they can stain for CD8+ T cells, dendritic cells and others as a follow up study?

This is a very interesting point, however, the main body of data suggests a direct effect of the alpha1-oleate complex on tumor cells, leading to programmed cell death. Reanalyzing the entire data set, we did not find evidence of immuno-modulatory effects on adaptive immunity. Innate immune response pathways were down regulated especially granulocyte response genes. A comment has been added to the revised discussion.

Reviewer #4 (Remarks to the Author): The revisions to the manuscript adequately address my concerns.

Reviewer #5 (Remarks to the Author): The manuscript is very well presented and the study conducted covered a broad range of biophysical characterisation, molecular dynamic simulation, cell assays and clinical work on the investigational drug: peptide-lipid complex. The authors have rightly noted its limited use in the presence of lipid binding proteins and hence the use in local administration in this case intravesical administration in NMIBC.

I am happy to recommend the manuscript to be accepted as it is.

Reviewer #6 (Remarks to the Author):

Review of "A novel, peptide-based approach for targeting and killing tumor cells with greater precision"

This work presents a comprehensive set of outcome data from the clinical trial “A First-in-Human Study of alpha1H in Patients With Non-muscle Invasive Bladder Cancer” with three primary endpoints: 1) Incidence of adverse events and classification in terms of severity, causality and outcome; 2) Change in cell shedding into urine and 3) Bladder tumors will be characterized by in vivo imaging during examination by cystoscopy.

The primary end points were

- *Safety analysis of alpha1-oleate treatment*
- *Tumor cell shedding*
- *Effects on tumor size*

These end points have been evaluated, using quantitative methods.

A. Safety was evaluated as the frequency of adverse events and effects on healthy bladder tissue.

Strict comparisons between the treatment- and placebo group show no evidence of drug-related adverse events. Furthermore, there was no evidence of a toxic response in healthy tissue biopsies from patients treated with alpha1-oleate, defined by histopathology or TUNEL staining.

This has been clarified in the revised Results section.

B. Alpha1-oleate triggered a rapid increase in cell shedding in all treated patients, at all visits.

The effect is shown by pairwise analysis of pre-and post-inoculation samples in individual patients as well as group-wise analysis comparing the treatment and placebo groups.

C. Effects on tumor size. *A significant reduction in lesion size was recorded in a blinded analysis performed by an independent expert not involved in the study.*

My comments are as follows:

1) Given that this study was designed as a randomized phase II trial a sample size justification of n=20 per arm should be presented in terms of the statistically detectable effect sizes for each of the three primary aims, i.e. no sample size rational was presented.

The sample size was based on a previous open study of HAMLET instillations in bladder cancer patients, where 8/9 patients showed a marked increase in cell shedding. Furthermore, tumor progression was inhibited in 100% of treated mice compared to the sham group, in the murine bladder cancer model. A sample size of 20 was therefore deemed suitable.

This point has been clarified in the Methods section of the revised manuscript.

2) A statistical comparison between Alpha1-o and Placebo should be carried out given the safety profiles were a primary endpoint. Currently only the descriptive counts are provided in Figure 4c.

The statistical analysis was presented in Figure 4c. As we did not detect any treatment-associated effects, drug-specific AEs cannot be reported. The Results and Figure legends have been revised.

3) There were several analyses where there were the factors of treatment and time. The proposed statistical analysis appears to be a two-way ANOVA when in fact it should be a repeated measures ANOVA to account for within subject variation, i.e. a factor for time, treatment and a treatment by time interaction term accounting for within subject correlation.

We have implemented the statistical analysis suggested by the reviewer and the results have been added to the figures where kinetic data is shown. Using repeated measures ANOVA did not change the interpretation of the data.

4) There appears to be missing data in several cases, eg the analysis corresponding to Figure 5i with the tumor size score and the analysis corresponding to Figure 6c. An explanation of missing data should be provided.

The tumor size score in Fig.5i included all cases where paired data was available (n=39). The sample numbers for Fig. 6c were 40 tumors in total and 38 healthy biopsies, as healthy biopsy were missing from two patients.

5) Are both treatment and control subject data contained in Figure 6d. If so might it be worthwhile to highlight the treatment arm, particularly if there are differential correlations.

Yes, both are included. Open circles indicate the placebo group in the revised Figure.

6) What were the statistical tests used for figure 1h

The data in Figure 1h was analyzed by paired t-tests. This has been explained in the revised legend to Figure 1.

7) The statistical plan mentions student's t-tests and the Wilcoxon signed rank test. However, neither of these methods were utilized and hence are not relevant to the current manuscript.

The t-test was used in Figure 1. The Wilcoxon sign rank test was used in a previous version of the manuscript.

REVIEWER COMMENTS

Reviewer #6 (Remarks to the Author):

The author's were responsive to the previous critique with the exception of the sample size justification. Stating that "A sample size of 20 patients per group was therefore deemed suitable" based on previous data is not a scientific rational.

Given the sample size is fixed then one should provide an effect size calculation, e.g. what is the detectable effect between the two groups given $\alpha=0.05$ and $\text{power}=0.80$. Without this information the actual interpretation of the results in terms of statistical significance (or not) is less clear. A quick call to a statistician would provide a solution to this issue, which is a standard part of any statistical methods section.

Response to Reviewer 6

A formal statistical analysis plan was submitted to and approved by the regulatory authorities.

Population variables, adverse events and efficacy variables were formally evaluated.

The Sample size calculation text from the clinical protocol is shown below.

Determination of Sample Size

The primary objective of the trial was to evaluate the safety of alpha1-oleate. No formal sample size calculation evaluating the power of the trial has been performed. However, a consideration regarding the sample size was made based on a previous open study of HAMLET instillations in bladder cancer patients and in the murine bladder cancer model. For efficacy, the sample size was based on analysis of change in tumor cells assessed before HAMLET instillation and after 2 hours. The mean fold increase of shed cells was 41.3 and standard deviation was 60.4 in 9 patients. A sample size of 20 patients per group was deemed suitable to achieve criterion for significance (alpha) 0.05 and power 90% using the paired samples 1-tailed t-test. The null hypothesis is H_0 : mean change in cell shedding = 0 and the alternative hypothesis is H_A : mean change in cell shedding > 0.

This information has been added to the revised Methods section.

The statistical analysis paragraph was further revised as shown below.

Statistical analysis

For efficacy, the sample size was based on analysis of change in tumor cells assessed from a previous study¹⁶. A sample size of 20 patients per group was deemed suitable to achieve criterion for significance (alpha) 0.05 and power 90% using the paired samples 1-tailed t-test. The null hypothesis is H_0 : mean change in cell shedding = 0 and the alternative hypothesis is H_A : mean change in cell shedding > 0. The Gaussian distribution was determined by the D'agostino & Pearson normality test. For data following a Gaussian distribution, student t-tests were used. Other data sets were analyzed by Mann-Whitney U-test. Correlations were determined by Spearman correlation. Kinetic data was analyzed using the repeated measures 2-way ANOVA test. All statistical analysis was done by using Prism version 6.02 (GraphPad Software Inc.). P values < 0.05 were considered as statistically significant. All images were created by the study team.